# Mitochondrial-cytochrome c oxidase II promotes glutaminolysis to sustain tumor cell survival upon glucose deprivation

Yong Yi [1] ✉, Guoqiang Wang[1], Wenhua Zhang[1], Shuhan Yu[1,2], Junjie Fei[1], Tingting An[1], Jianqiao Yi[1], Fengtian Li[3], Ting Huang[2], Jian Yang[1], Mengmeng Niu [1], Yang Wang [1] ✉, Chuan Xu[2] ✉ & Zhi-Xiong Jim Xiao [1,2,4] ✉

Glucose deprivation, a hallmark of the tumor microenvironment, compels tumor cells to seek alternative energy sources for survival and growth. Here, we show that glucose deprivation upregulates the expression of mitochondrial-cytochrome c oxidase II (MT-CO2), a subunit essential for the respiratory chain complex IV, in facilitating glutaminolysis and sustaining tumor cell survival. Mechanistically, glucose deprivation activates Ras signaling to enhance *MT-CO2* transcription and inhibits IGF2BP3, an RNA-binding protein, to stabilize MT-CO2 mRNA. Elevated MT-CO2 increases flavin adenosine dinucleotide (FAD) levels in activating lysine-specific demethylase 1 (LSD1) to epigenetically upregulate *JUN* transcription, consequently promoting glutaminase-1 (GLS1) and glutaminolysis for tumor cell survival. Furthermore, MT-CO2 is indispensable for oncogenic Ras-induced glutaminolysis and tumor growth, and elevated expression of MT-CO2 is associated with poor prognosis in lung cancer patients. Together, these findings reveal a role for MT-CO2 in adapting to metabolic stress and highlight MT-CO2 as a putative therapeutic target for Ras-driven cancers.

Metabolic reprogramming is a hallmark of cancer cells[1]. Most cancer cells use glucose as the major energy source and preferentially catabolize glucose to lactate even under abundant oxygen conditions, a phenomenon termed the Warburg effect or aerobic glycolysis[1]. Aerobic glycolysis can rapidly but inefficiently generate ATP by consuming large amounts of glucose to meet the needs of rapidly proliferating tumor cells[1,2]. It has been well-known that glucose deprivation is one of the characteristics of the tumor microenvironment, which can be attributed to the large consumption of glucose by tumor cells[1,2], and by insufficient blood supply due to poor angiogenesis[3]. Indeed, it has been reported that the glucose concentrations in the solid tumor tissues are 3- to 10-fold lower than in normal tissues[4,5]. Therefore, Tumor

cells must sense glucose shortage and seek alternative energy resources for tumor cell survival and proliferation.

Glutamine is an alternative source of energy for cancer cells. Glutamine enters the mitochondria and is catabolized to glutamate by glutaminase-1 (GLS1)[6]. Glutamate can then be converted by glutamate dehydrogenase (GLUD1) to α-ketoglutarate (α-KG), which then enters the TCA cycle and is ultimately oxidized to produce ATP[7]. Furthermore, glutamine-derived α-KG can be reductively carboxylated to generate citrate, which can generate acetyl-CoA for de novo fatty acid synthesis[8]. Moreover, glutamine-derived glutamate also can be converted to non-essential amino acid (NEAA) by alanine or aspartate transaminases (TAs) for protein synthesis[9]. It is reported that

[1]Center of Growth, Metabolism and Aging, Key Laboratory of Bio-Resource and Eco-Environment, Ministry of Education, College of Life Sciences, Sichuan University, Chengdu, China. [2]Department of Oncology & Cancer Institute, Department of Laboratory Medicine and Sichuan Provincial Key Laboratory for Human Disease Gene Study, Sichuan Academy of Medical Sciences, Sichuan Provincial People's Hospital, University of Electronic Science and Technology of China, Chengdu, China. [3]School of Biosciences and Technology, Chengdu Medical College, Chengdu, China. [4]State Key Laboratory of Biotherapy, West China Hospital, Sichuan University, Chengdu, China. ✉e-mail: yy-yiyong@scu.edu.cn; wangy90@scu.edu.cn; xuchuan100@163.com; jimzx@scu.edu.cn

glutamine-derived TCA cycle intermediates, including fumarate, malate, and citrate, are significantly increased upon glucose deprivation[10]. However, how glucose deprivation triggers metabolic reprogramming to the glutaminolysis pathway for tumor cell survival and growth remains unclear.

Mitochondria are intracellular organelles with the core function to produce the majority of ATP by oxidative phosphorylation[11]. In mammalian cells, mitochondria consist of 16,569 base pairs of DNA organized in a closed circle, encoding 13 proteins, which are core components of the respiratory chain[12]. It has been shown that cancer cells lacking mitochondrial DNA are sensitive to glucose deprivation[13], underlying a critical role for mitochondria in cellular response to glucose deprivation. However, how mitochondria act in the cellular response to glucose deprivation is unknown.

In this study, we show that glucose deprivation leads to robust upregulation of the expression of MT-CO2, a mitochondrial genome-encoded protein essential in the complex IV of the respiratory chain, which bridges Ras signaling to epigenetic upregulation of *JUN* gene transcription via activation of the FAD-LSD1-H3K9me2 axis, consequently resulting in elevated GLS1 expression, glutaminolysis, and tumor cell survival. Inhibition of MT-CO2 by shRNA or berberine leads to inhibition of glutaminolysis and suppression of Ras-driven tumor growth.

## Results

### Mitochondrial MT-CO2 is essential for cancer cell survival upon glucose deprivation and for tumor growth in vivo

Glucose deprivation is often associated with tumor development. However, glucose deprivation can lead to an energy crisis and cell death. Thus, cancer cells must adopt alternative energy sources for cell survival and proliferation. To investigate the underlying molecular changes that lead to cell survival upon glucose deprivation, we selected human lung cancer H292 cells that could survive in DMEM lacking glucose but supplemented with 2 mM glutamine (Glc−/Gln+) and obtained two sets of viable H292 cells (H292-V#1 and H292-V#2) (Supplementary Fig. S1a). As shown in Supplementary Fig. S1b–d, while parental H292 cells were inviable in the Glc-/Gln+ condition, H292-V cells were well survived, as evidenced by either trypan blue exclusion assay or propidium iodide (PI) staining assay. Notably, xenograft tumor mouse models showed that H292-V cells exhibited higher tumorigenicity than parental H292 cells (Fig. 1a–c and Supplementary Fig. S1e, f), indicating that H292-V cells acquired a growth advantage compared to parental H292 cells. Notably, as shown in Supplementary Fig. S1g, h, ethidium bromide (EB)-induced depletion of mitochondrial DNA significantly increased the sensitivity of human lung cancer A549 cells to glucose deprivation, in keeping with a previous report[13]. We, therefore, hypothesized that mitochondrial genome-encoded proteins play a critical role in cancer cell survival upon energy stress. We thus examined the expression of genes encoded by the mitochondrial genome in H292-V and parental H292 cells. As shown in Fig. 1d, compared to parental H292 cells, the steady-state mRNA levels of *MT-CO2*, which encodes a core subunit of respiratory chain complex IV, were dramatically increased in H292-V cells. Since there was little difference in mitochondrial biogenesis between H292-V and parental H292 cells, as evidenced by the comparable ratio of mitochondrial DNA/ nuclear DNA or citrate synthase levels (Supplementary Fig. S1i, j), it suggests that the robust elevation of the MT-CO2 mRNA levels is most likely attributed to the increased gene transcription and/or increased mRNA stability. Consistent with this notion, western blot analyses showed that H292-V cells had higher MT-CO2 protein expression than parental H292 cells (Fig. 1e).

Next, we investigated the biological significance of elevated mitochondrial MT-CO2 in sustained tumor cell survival upon glucose starvation. As shown in Supplementary Fig. S1k, l, A549 or H1299 cells with high levels of MT-CO2 survived well upon glucose deprivation. By contrast, H292 or H1975 cells, both of which had low MT-CO2

expression, were unable to survive (Supplementary Fig. S1k, l). To inhibit MT-CO2 expression in A549 and H1299 cells, we designed and verified two different shRNAs that can effectively knockdown MT-CO2 expression (Fig. 1f), in keeping with a previous report that RNAi technology can be used to silence the mitochondrial genome-encoding genes in vitro[14]. As shown in Fig. 1g, silencing of MT-CO2 dramatically inhibited A549 and H1299 cell viability upon glucose deprivation. In addition, the knockdown of MT-CO2 also significantly inhibited the proliferation of A549 and H1299 cells (Fig. 1h, i), indicating that MT-CO2 is essential for tumor cell growth even in an environment rich in glucose and glutamine. Consistently, in vivo xenograft tumor mouse models showed that knockdown of MT-CO2 dramatically inhibited tumor growth, concomitant with reduced Ki67 expression (Fig. 1j–l and Supplementary Fig. S1m–o). Importantly, the silencing of MT-CO2 significantly re-sensitized H292-V cells to glucose deprivation (Fig. 1m, n). Moreover, the knockdown of MT-CO2 in H292-V cells markedly inhibited cell proliferation in vitro and tumor growth in vivo (Fig. 1o–r and Supplementary Fig. S1p, q).

Taken together, these results indicate that glucose deprivation leads to upregulation of MT-CO2, which plays a pivotal role in cancer cell survival upon glucose deprivation and in tumor growth in vivo.

### MT-CO2 elevates GLS1 expression in promoting glutaminolysis critical for cancer cell survival upon glucose deprivation

Next, we investigated the molecular basis by which MT-CO2 regulates cancer cell survival upon glucose deprivation. Glutamine, fatty acids, or glycogen can serve as alternative energy sources for cancer cells[15–17]. Notably, inhibition of fatty acid oxidation by etomoxir, a selective inhibitor of carnitine palmitoyl-transferase 1a (CPT1a), or inhibition of glycogen catabolism by CP-91149, a glycogen phosphorylase inhibitor, led to upregulated expression of p53 and p21 (Supplementary Fig. S2a), as previously reported[18,19]. However, either etomoxir or CP-91149 had little effect on the viability of A549 or H1299 cells under Glc−/Gln+ condition (DMEM containing 2 mM glutamine in the absence of glucose) (Fig. 2a, b and Supplementary Fig. S2b, c). By sharp contrast, inhibition of glutaminolysis by CB-839, a selective inhibitor of glutaminase-1 (GLS1), dramatically suppressed the survival of A549 and H1299 cells under Glc-/Gln+ condition (Fig. 2a, b and Supplementary Fig. S2b, c), indicating that cancer cells preferentially utilize glutamine as an energy source for survival upon glucose deprivation. Consistently, depletion of glutamine resulted in robust cancer cell death in the absence of glucose (Supplementary Fig. S2d). Furthermore, ectopic expression of GLS1 in H292 cells markedly upregulated cellular oxygen consumption rate (OCR), cell survival under the Glc-/Gln+ condition, and tumor growth in vivo (Supplementary Fig. S2e–i). These results indicate that glutamine is the preferred alternative energy source and that glutaminolysis is essential for cancer cell survival under glucose deprivation.

Given that glutaminolysis plays a pivotal role in maintaining tumor cell survival upon glucose deprivation, we hypothesized that MT-CO2 could be involved in regulating glutaminolysis. To address this issue, we performed mass spectrometry analyses on energy metabolites. As shown in Supplementary Fig. S2j, the silencing of MT-CO2 in H1299 cells dramatically increased cellular glutamine levels. Notably, the expression of glutamine transporter, SLC1A5, was not affected by knockdown of MT-CO2 (Supplementary Fig. S2k), suggesting that silencing of MT-CO2 may lead to blockage of glutaminolysis which leads to cellular glutamine accumulation. We then traced the metabolic flux of $^{13}$C-labeled glutamine ($^{13}C_5$-glutamine). As shown in Fig. 2c, d, there was a reduced enrichment of $^{13}$C from $^{13}C_5$-glutamine in downstream metabolites such as glutamate, α-ketoglutarate, succinate, fumarate, and citrate. Additionally, $^{13}C_5$-glutamine accumulation was observed upon knockdown of MT-CO2 (Fig. 2d), indicating that silencing MT-CO2 inhibits glutaminolysis, blocking tumor cells from utilizing glutamine as a carbon source.

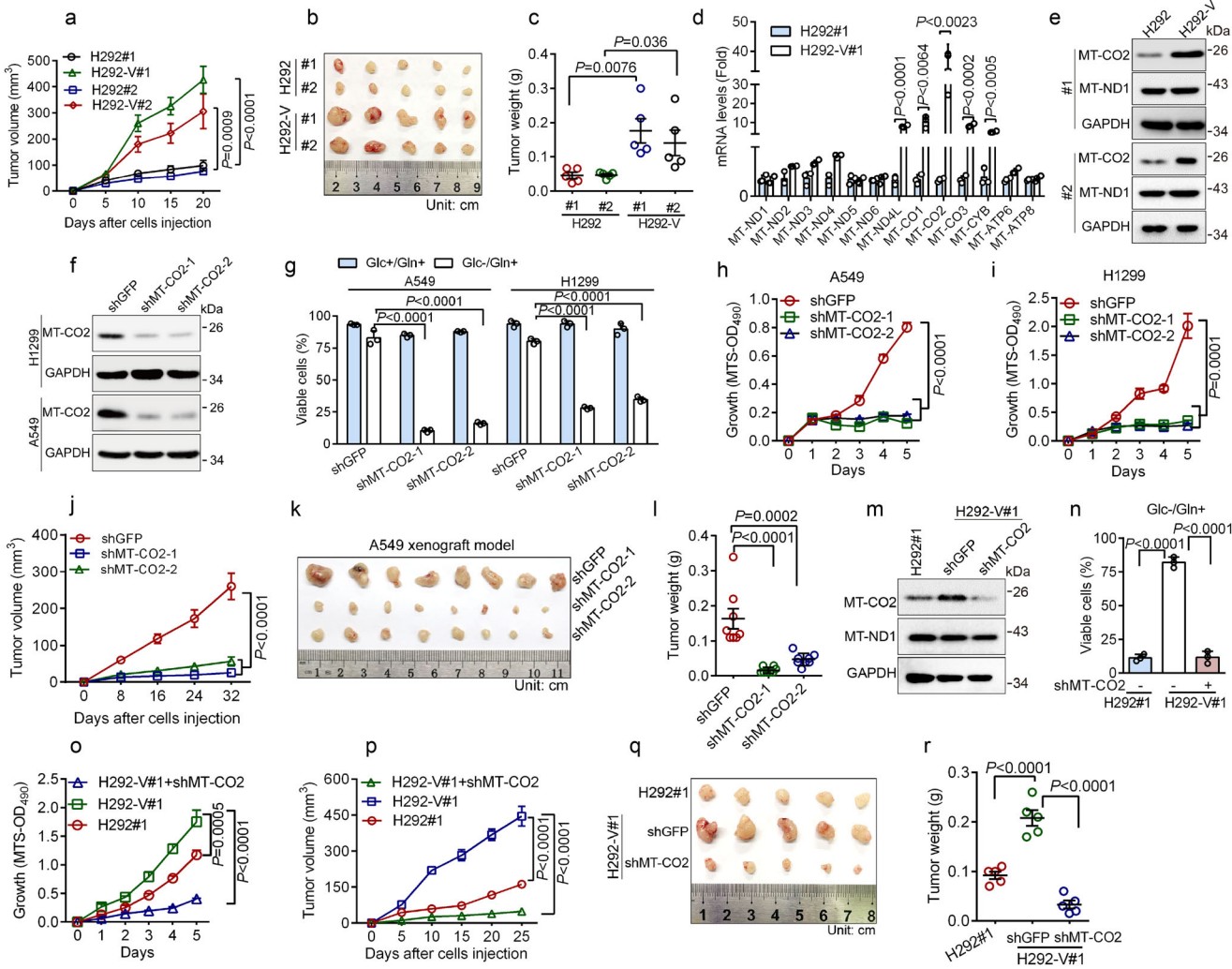

**Fig. 1 | MT-CO2 is essential for tumor cell survival and tumor growth upon glucose deprivation. a–c** H292 or glucose-starvation-resistant H292 (H292-V) cells ($5 \times 10^5$) were subcutaneously inoculated into the right scruff of the female BALB/c nude mouse ($n = 5$/group). The tumor volumes on indicated days after injection were examined (**a**). Twenty days post-injection, tumors were weighed (**c**), and photos were taken (**b**). H292 or H292-V cells were subjected to qPCR analyses (**d**, $n = 3$ independent experiments) or were subjected to western blot analyses (**e**). **f, g** H1299 or A549 cells stably expressing shGFP, shMT-CO2-1, or shMT-CO2-2 were subjected to western blot analyses (**f**) or were grown in DMEM containing 2 mM glutamine in the absence of glucose (Glc−/Gln+) for 24 h. Cell viability was determined by trypan blue exclusion assay (**g**, $n = 3$ independent experiments). **h–l** H1299 or A549 cells stably expressing shGFP, shMT-CO2–1, or shMT-CO2-2 were subjected to MTS analyses for cell growth (**h**, **i**, $n = 3$ independent experiments). For tumor growth, indicated cells ($1 \times 10^6$) were subcutaneously inoculated into the right scruff of the female BALB/c nude mouse ($n = 8$/group). The tumor volumes on indicated days after injection were examined (**j**). Thirty-two days post-injection, tumors were weighed (**l**) and photos were taken (**k**). **m–r** H292-V cells stably expressing shGFP or shMT-CO2 were subjected to western blot analyses (**m**) or were grown in Glc−/Gln+ condition for 48 h followed by trypan blue exclusion assay for cell viability (**n**, $n = 3$ independent experiments) or were subjected to MTS analyses for cell growth (**o**, $n = 3$ independent experiments). For tumor growth, cells ($5 \times 10^5$) were subcutaneously inoculated into the right scruff of the female BALB/c nude mouse ($n = 5$/group). The tumor volumes on indicated days after injection were examined (**p**). Twenty-five days post-injection, tumors were weighed (**r**) and photos were taken (**q**). These experiments have been repeated three times with similar results (**e**, **f**). Data were presented as mean ± SD (**d**, **g**, **h**, **i**, **n**, **o**) or SEM (**a**, **c**, **j**, **l**, **p**, **r**). Comparisons were performed with one-way ANOVA with Tukey's test (**g**, **l**, **n**, **r**), two-way ANOVA with Tukey's test (**a**, **h–j**, **o**, **p**), and unpaired two-tailed Student's *t* test (**c**, **d**).

To investigate the molecular basis by which MT-CO2 regulates glutaminolysis, we examined the effects of silencing of MT-CO2 on the expression of the glutaminolysis-related enzymes. As shown in Fig. 2e–g, silencing of MT-CO2 significantly inhibited the mRNA and protein expression of GLS1, while it had little effect on the expression of glutamate dehydrogenase (GLUD1), which catalyzes glutamate to α-ketoglutarate (α-KG). Importantly, silencing of MT-CO2-mediated inhibition of cell viability under the Glc-/Gln+ condition was significantly rescued by supplement of dimethyl α-KG (DM-αKG, a cell-permeable analog of α-KG) (Fig. 2h, i) or ectopic expression of GLS1 (Fig. 2j, k). Moreover, ectopic expression of GLS1 also significantly rescued tumor growth in vivo inhibited by silencing of MT-CO2 (Fig. 2l–r).

Together, these results indicate that silencing of MT-CO2 inhibits glutaminolysis and tumor growth via the downregulation of GLS1 expression.

## MT-CO2 facilitates *JUN* transcription through activation of the FAD-LSD1-H3K9me2 axis to promote GLS1 expression

To explore the molecular mechanism(s) by which MT-CO2 regulates GLS1 expression, we performed RNA-seq analyses. As shown in Fig. 3a, the silencing of MT-CO2 significantly affected the expression of a subset of transcription factors, including c-JUN, KLF2, FOXM1, and SOX2. Notably, c-JUN has been reported as a transcription factor of GLS1[20], raising the possibility for MT-CO2 regulating GLS-1 through

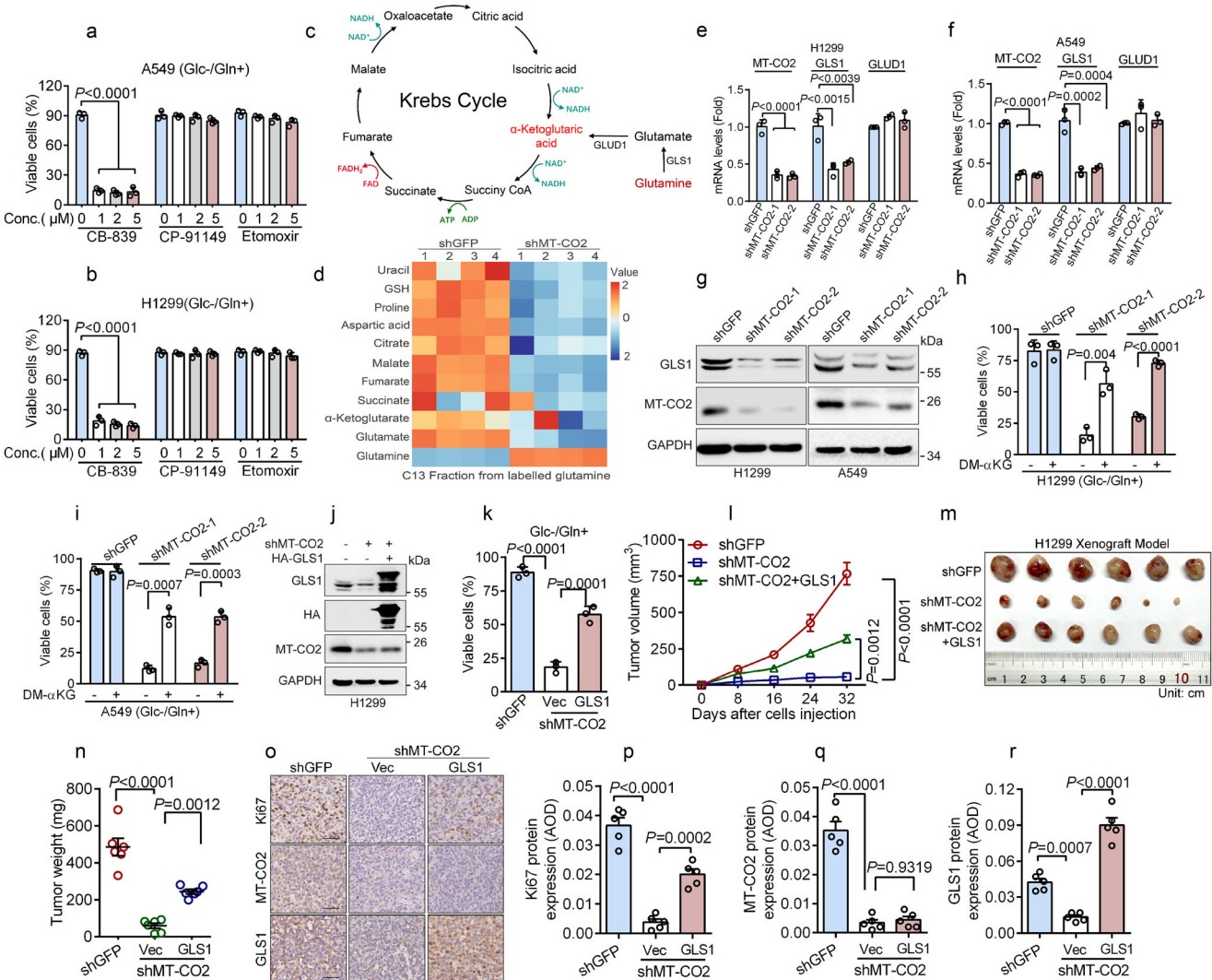

**Fig. 2 | MT-CO2 elevates GLS1 expression in facilitating glutaminolysis to promote tumor cell survival upon glucose deprivation. a, b** A549 or H1299 cells grown in DMEM containing 2 mM glutamine in the absence of glucose (Glc−/Gln+) were treated with as indicated inhibitors for 36 h (A549) or 48 h (H1299). Cell viability was determined by trypan blue exclusion assay (*n* = 3 independent experiments). **c** A sketch depicts the process of glutaminolysis. **d** H1299 cells stably expressing shGFP or shMT-CO2 were cultured in DMEM containing 2 mM $^{13}C_5$-glutamine for 8 h. Cells were subjected to mass isotopomer analyses (*n* = 4 biologically independent samples per experiment). **e–i** H1299 or A549 cells stably expressing shGFP, shMT-CO2-1, or shMT-CO2-2 were subjected to qPCR (**e, f**, *n* = 3 independent experiments) or western blot (**g**) analyses or cells were grown in Glc−/Gln+ with or without supplement of 500 μM dimethyl α-Ketoglutaric acid (DM-αKG) for 24 h. Cell viability was determined by trypan blue exclusion assay (**h, i**, *n* = 3 independent experiments). **j, k** H1299-shMT-CO2-1cells stably expressing HA-GLS1 were subjected to western blot analyses (**j**) or cells were grown in Glc−/

Gln+ condition for 24 h. Cell viability was determined by trypan blue exclusion assay (**k**, *n* = 3 independent experiments). The samples derive from the same experiment but different gels for GLS1, MT-CO2, GAPDH, and another for HA were processed in parallel (**j**). **l–r** Cells (1 × 10⁶) were subcutaneously inoculated into the right scruff of the female BALB/c nude mouse (*n* = 6/group). The tumor volumes on indicated days after injection were examined (**l**). Thirty-two days post-injection, tumors were weighed (**n**) and photos were taken (**m**). The tumor samples were subjected to immunohistochemistry staining analyses for Ki67, MT-CO2, or GLS1 expression (**o**). The protein levels were quantified by average optical density (AOD) and were presented (**p–r**). Scale bar, 50 μm. These experiments have been repeated three times with similar results (**g, j**). Data were presented as mean ± SD (**a, b, e, f, h, i, k**) or SEM (**l, n, p–r**). Comparisons were performed with one-way ANOVA with Tukey's test (**a, b, e, f, k, n, p–r**), two-way ANOVA with Tukey's test (**l**), and unpaired two-tailed Student's *t* test (**h, i**).

c-JUN. Indeed, similar to the effects of silencing MT-CO2, the viability of H1299 cells was also dramatically suppressed by the knockdown of c-JUN under Glc−/Gln+ condition (DMEM containing 2 mM glutamine in the absence of glucose) (Supplementary Fig. S3a, b). Furthermore, qPCR and western blot analyses showed that silencing of MT-CO2 significantly inhibited c-JUN mRNA and protein levels (Fig. 3b, c). Importantly, the silencing of MT-CO2-mediated downregulation of GLS1 expression was completely rescued by ectopic expression of c-JUN (Fig. 3d−e). Together, these results indicate that silencing of MT-CO2 leads to inhibition of *GLS1* transcription via suppression of c-JUN expression.

We then investigated the molecular basis by which MT-CO2 regulates *JUN* transcription. Accumulating evidence indicates that disruption of mitochondria homeostasis leads to the alteration of a set of metabolites that can serve as signaling molecules, including AMP/ATP, ROS, NAD+, and acetyl coenzyme A, which are involved in the regulation of gene transcription through various mechanisms[21]. Therefore, we speculated that silencing of MT-CO2 may affect certain signaling molecules to regulate c-JUN expression. Notably, treatment with NAC (N-Acetyl-L-cysteine, a ROS scavenger), nicotinamide mononucleotide (NMN, an NAD+ precursor), ATP, or NaAc (Sodium acetate) had little effect on the silencing of MT-CO2-mediated

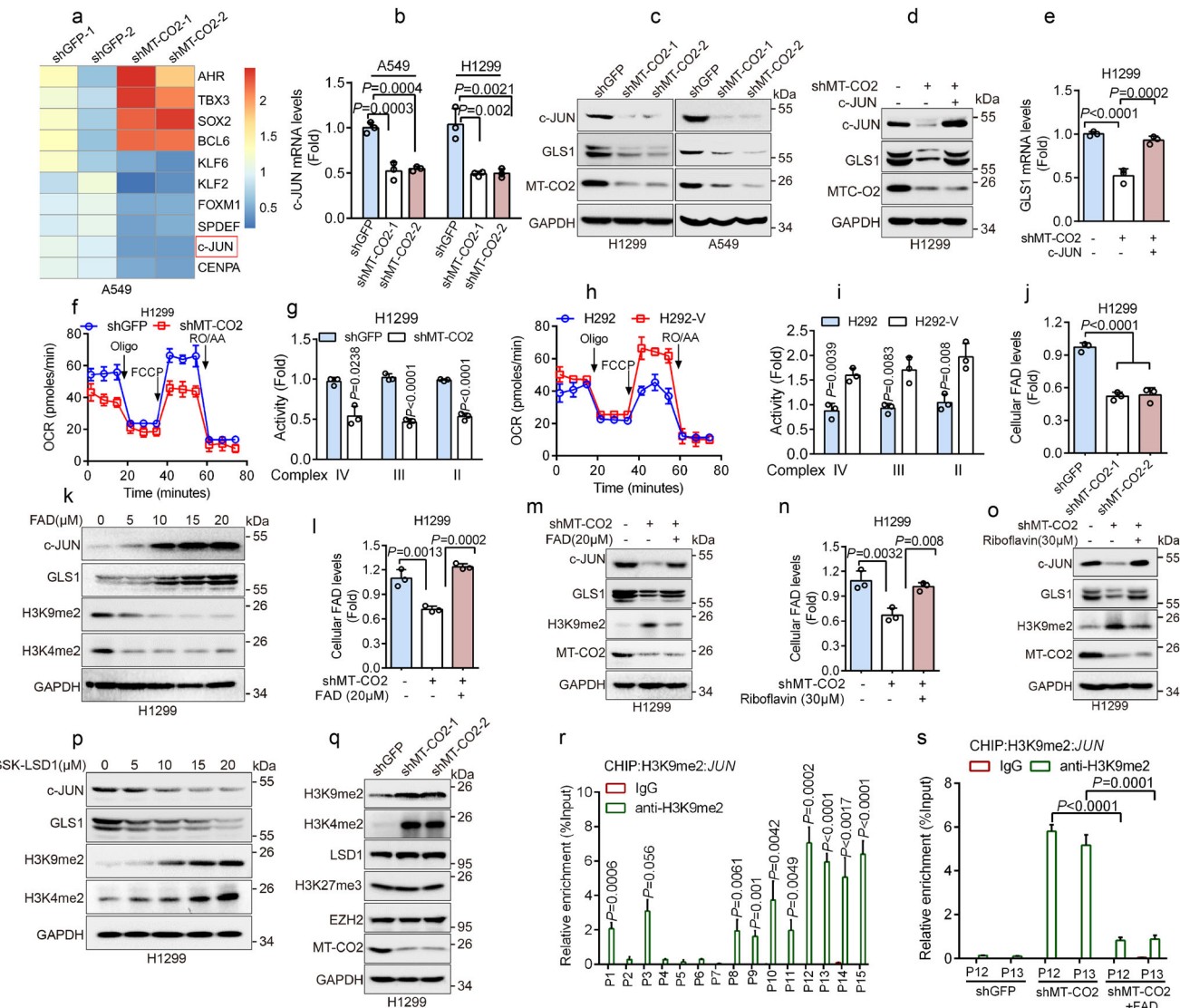

**Fig. 3 | MT-CO2 facilitates *JUN* transcription through the FAD-LSD1-H3K9me2 axis to promote GLS1 expression. a–c** Indicated cells were subjected to RNA-seq (**a**, *n* = 2 biologically independent samples per experiment), qPCR (**b**, *n* = 3 independent experiments), or western blot (**c**) analyses. **d**, **e** H1299-shMT-CO2 cells were transfected with c-JUN for 48 h, followed by western blot (**d**) or qPCR (**e**, *n* = 3 independent experiments) analyses. The samples derive from the same experiment but different gels for GLS1, MT-CO2, GAPDH, and another for c-JUN were processed in parallel (**c**, **d**). **f–i** Indicated cells were subjected to measurement of the oxygen consumption rate (OCR) (**f**, **h**) or complex (II, III, IV) activity (**g**, **i**) (*n* = 3 independent experiments). **j** Indicated cells were subjected to examine cellular FAD levels (*n* = 3 independent experiments). **k** H1299 cells were treated with an indicated concentration of FAD for 48 h, followed by western blot analyses. **l–o** Indicated cells were treated with an indicated concentration FAD or riboflavin for 48 h, followed by examining the cellular FAD levels (**l**, **n**, *n* = 3 independent experiments) or western blot analyses (**m**, **o**). The samples derive from the same experiment but different gels for GLS1, H3K9me2, GAPDH, and another for c-JUN, MT-CO2 were processed in parallel (**m**, **o**). **p** H1299 cells were treated with an indicated concentration of GSK-LSD1 for 36 h, followed by western blot analyses. The samples derive from the same experiment but different gels for GLS1, H3K9me2, GAPDH, and another for c-JUN, H3K4me2 were processed in parallel (**k**, **p**). **q** Indicated cells were subjected to western blot analyses. The samples derive from the same experiment but different gels for LSD1, H3K9me2, GAPDH, another for EZH2, H3K27me3, another for MT-CO2, and another for H3K4me2 were processed in parallel. **r**, **s** H1299 cells were treated with 20 μM GSK-LSD1 for 36 h (**r**) or H1299-shMT-CO2 cells were treated with 20 μM FAD for 48 h (**s**), followed by CHIP-qPCR analyses (*n* = 3 independent experiments). These experiments have been repeated three times with similar results (**c**, **d**, **k**, **m**, **o**–**q**). Data were presented as mean ± SD (**b**, **e**, **f–j**, **l**, **n**, **r**, **s**). Comparisons were performed with one-way ANOVA with Tukey's test (**b**, **e**, **j**, **l**, **n**) and unpaired two-tailed Student's *t* test (**g**, **i**, **r**, **s**).

downregulation of c-JUN and GLS1 expression (Supplementary Fig. S3c–f).

Since MT-CO2 is a key component in the complex IV of the respiratory chain, we examined whether disruption of electron transport chain function would affect c-JUN and GLS1 expression. As shown in Supplementary Fig. S4a, b, inhibition of respiratory chain complex I by small molecule inhibitors, such as rotenone and metformin, did not affect c-JUN expression. In addition, while silencing of MT-ND1 (a subunit of respiratory chain complex I) suppressed the cellular oxygen consumption rate (OCR) as expected, it had little

effect on c-JUN and GLS1 expression (Supplementary Fig. S4g, j), suggesting that the disruption of complex I is not engaged in the regulation of c-JUN/GLS1. In contrast, inhibition of respiratory chain complexes II, III, IV, and V by the respective inhibitors (Carboxin for complex II, Antimycin for complex III, $NaN_3$ for complex IV, and Oligomycin for complex V) significantly suppressed c-JUN expression (Supplementary Fig. S4c–f), suggesting that complex II-V of the respiratory chain is important in the regulation of c-JUN. Notably, the knockdown of MT-CO2 of complex IV not only inhibited expression of c-JUN and GLS1, but also significantly suppressed the

OCR levels and the activity of complex II, III, and IV (Fig. 3f, g), in keeping with the observation that glucose-starvation-resistant H292 cells (H292-V) exhibited higher MT-CO2 and OCR levels, along with increased activity of complex II, III, and IV (Figs. 1e, 3h, i). Consistently, the knockdown of MT-CYB (a subunit of respiratory chain complex III) or MT-ATP6 (a subunit of respiratory chain complex V) also led to a marked suppression of OCR levels as well as c-JUN and GLS1 expression (Supplementary Fig. S4h, i, S4k, l). These results suggest that the complex II-V of the respiratory chain plays an important role in regulating the expression of c-JUN and GLS1. It has been documented that the complex II-V of the respiratory chain is essential for FADH$_2$-mediated ATP generation and flavin adenosine dinucleotide (FAD) regeneration[22–25]. Thus, it would be expected that the disruption of complex II-V of the respiratory chain should result in a reduction of FAD levels. Indeed, the knockdown of MT-CO2, MT-CYB, or MT-ATP6 significantly reduced cellular FAD levels (Fig. 3j and Supplementary Fig. S4m).

To explore whether FAD can rescue the downregulation of c-JUN and GLS1 expression mediated by silencing MT-CO2, we examined the effects of FAD and riboflavin, the precursor of FAD, on intracellular FAD levels. As shown in Supplementary Fig. S4n, o, treatment with FAD or riboflavin significantly increased intracellular FAD levels in a dose-dependent manner, in keeping with a previous report[26]. Notably, FAD treatment led to a marked upregulation of c-JUN and GLS1 in a time- and dose-dependent manner (Fig. 3k and Supplementary Fig. S4p). Importantly, the silencing of MT-CO2-mediated downregulation of c-JUN and GLS1 protein expression could be effectively rescued by FAD or riboflavin (Fig. 3l–o).

It has been documented that FAD is a key cofactor of lysine-specific histone demethylase 1 (LSD1, also known as KDM1A), which catalyzes the demethylation of histone 3, including H3K4me2 and H3K9me2, to regulate gene expression[27]. Therefore, we speculated that the MT-CO2-FAD axis might be involved in epigenetically regulating c-JUN expression via LSD1. To address this issue, we examined the effects of LSD1 on the c-JUN protein. As shown in Fig. 3p and Supplementary Fig. S5a, treatment with GSK-LSD1, a specific inhibitor of LSD1, significantly inhibited c-JUN and GLS1 expression, concomitant with markedly upregulated H3K4me2 and H3K9me2 levels. Silencing of MT-CO2 also dramatically increased H3K9me2 and H3K4me2 levels, whereas it had little effect on H3K27me3 expression (Fig. 3q).

Consistent with the knowledge that H3K9me2 serves as a repressive epigenetic mark[28], silencing of MT-CO2 upregulated H3K9me2 levels and downregulated c-JUN expression, which was largely rescued by FAD or riboflavin treatment, but not by α-ketoglutarate (Fig. 3m, o, and Supplementary Fig. S5b). We further investigated whether H3K9me2 could bind to the *JUN* gene promoter to inhibit *JUN* transcription. To address this issue, we designed 15 primers (P1 to P15) that can cover the -1 to -4000 region of the *JUN* gene promoter (Supplementary Fig. S5c). Our CHIP-qPCR analyses indicated that H3K9me2 could strongly bind to P10 to P15 (-2405 to -4000) region of the *JUN* gene promoter (Fig. 3r). Silencing of MT-CO2 significantly increased H3K9me2 binding to the P12 and P13 regions of the *JUN* gene promoter, both of which could be largely rescued by FAD treatment (Fig. 3s). Parallel experiments showed that H3K4me2 could also bind to the *JUN* gene promoter, which was unaffected by the silencing of MT-CO2 (Supplementary Fig. S5d–f).

Together, these results indicate that the knockdown of MT-CO2 leads to reduced electron flows of complex II-IV and FAD levels, resulting in reduced LSD1 enzymatic activities, which in turn downregulates *JUN* transcription via the elevation of H3K9me2 levels, consequently resulting in suppressing GLS1 expression.

## Glucose deprivation activates Ras signaling to promote the MT-CO2-GLS1-glutaminolysis axis required for cancer cell survival and tumor growth

Our aforementioned data indicate that glucose deprivation promotes MT-CO2 expression to facilitate glutaminolysis and tumor cell survival. Yet, how glucose starvation upregulates MT-CO2 is unknown. To explore the underlying molecular mechanism, we performed RNA-seq analyses to profile gene expression in glucose-starvation-resistant H292 cells (H292-V) versus parental H292 cells. As shown in Supplementary Fig. S6a, Kyoto Encyclopedia of Genes and Genomes (KEGG) pathway analyses of DEGs (Differentially expressed genes) showed that multiple signaling pathways, including Ras signaling, MAPK signaling, Hippo signaling, and Wnt signaling were significantly enriched in H292-V cells. In addition, whole exon sequencing (WES) analyses showed that compared to H292 cells, H292-V cells had no acquisition of additional gene mutations in the Ras pathway (Supplementary Table 1). However, Gene Ontology (GO) enrichment analyses showed that, in H292-V cells, there was the acquisition of gene mutations enriched in multiple GO terms, which can affect the Ras signaling, such as Ras GTPase binding, Rab GTPase binding, and GTPase activator activity (Supplementary Fig. S6b). Therefore, we speculated that glucose deprivation could promote MT-CO2 expression via activating the Ras signaling. Indeed, western blot analyses showed that the Ras downstream effector, ERK, was significantly activated in H292-V cells (Fig. 4a).

We next examined the effects of activated Ras on MT-CO2 expression and tumor cell survival upon glucose starvation. As shown in Supplementary Fig. S1k, l, A549 or H1299 cells harboring activated Ras (KRas$^{G12S}$ or NRas$^{Q61K}$, respectively) could survive well under Glc-/Gln+ (DMEM containing 2 mM glutamine in the absence of glucose) condition, concomitant with higher MT-CO2 protein expression. By contrast, H292 or H1975 cells harboring wild-type Ras and expressing lower levels of MT-CO2 were unable to survive (Supplementary Fig. S1k, l). Notably, ectopic expression of KRas$^{G12V}$ in H292 or H1975 cells promoted cell survival under the Glc-/Gln+ condition, concomitant with dramatically elevated mRNA and protein levels of MT-CO2 (Fig. 4b–d). Furthermore, not only KRas$^{G12V}$ but also HRas$^{G12V}$ could significantly elevate MT-CO2 expression (Supplementary Fig. S6c). By contrast, silencing of KRas in A549 or H292-V cells significantly downregulated MT-CO2 protein expression (Fig. 4e, g), concomitant with inhibited A549 or H292-V cell viability under Glc-/Gln+ condition (Fig. 4f, h). Importantly, the knockdown of MT-CO2 completely inhibited KRas$^{G12V}$-induced survival of H292 or H1975 cells under the Glc-/Gln+ condition (Fig. 4i, j). Together, these results indicate that activation of Ras signaling promotes the upregulation of MT-CO2 expression and that MT-CO2 is indispensable to Ras-mediated cell viability in response to glucose deprivation.

It has been established that activation of Ras is a driving force of glutaminolysis through promoting the expression of glutamine transporter, SLC1A5, or glutamate-oxaloacetate transaminase (GOT)[8,29]. Whether Ras-mediated upregulation of MT-CO2 plays a role in Ras-dependent glutaminolysis is unknown. As shown in Supplementary Fig. S7a, b, consistent with glucose-starvation-resistant H292 cells, ectopic expression of KRas$^{G12V}$ significantly upregulated the activity of respiratory chain complex II-IV and oxygen consumption rate (OCR). Moreover, ectopic expression of KRas$^{G12V}$ also significantly upregulated c-JUN and GLS1 protein expression, FAD levels as well as dramatically downregulated H3K9me2 levels (Supplementary Fig. S7c–e), all of which could be completely rescued by either silencing of MT-CO2 or treatment with LSD1 inhibitor GSK-LSD1 (Supplementary Fig. S7f–i). Importantly, the knockdown of MT-CO2 inhibited KRas$^{G12V}$-induced cell survival under the Glc-/Gln+ condition, which could be markedly rescued by supplement of dimethyl α-KG (DM-αKG) (Supplementary Fig. S7j). Together, these results indicate that activation of Ras promotes glutaminolysis via activating the MT-CO2-FAD-LSD1-H3K9me2-c-JUN-GLS1 axis.

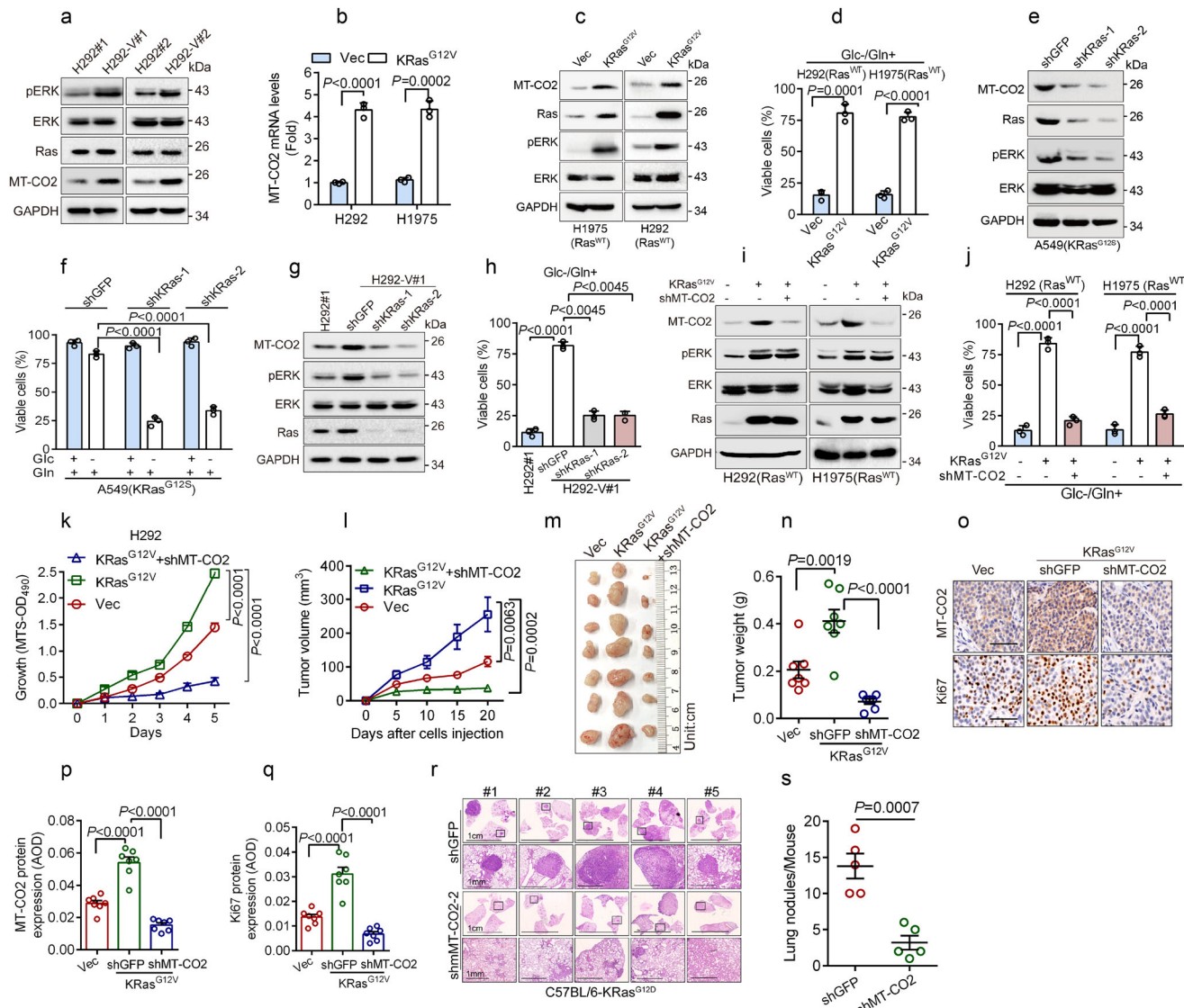

**Fig. 4 | Glucose deprivation activates Ras signaling to promote MT-CO2 expression critical for cancer cell survival and tumor growth. a** H292 or glucose-starvation-resistant H292 (H292-V) cells were subjected to western blot analyses. **b–d** H292 or H1975 cells expressing $KRas^{G12V}$ were subjected to qPCR (**b**, $n = 3$ independent experiments) or western blot (**c**) analyses or were grown in DMEM containing 2 mM glutamine in the absence of glucose (Glc−/Gln+) for 24 h, followed by examining cell viability (**d**, $n = 3$ independent experiments). **e–h** A549 or H292-V cells expressing shGFP, shKRas-1, or shKRas-2 were subjected to western blot analyses (**e, g**) or cells were grown in Glc−/Gln+ condition for 24 h (A549) or 48 h (H292-V), followed by examining cell viability (**f, h**, $n = 3$ independent experiments). **i, j** H292-$KRas^{G12V}$ or H1975-$KRas^{G12V}$ cells expressing shMT-CO2 were subjected to western blot analyses (**i**) or cells were grown in Glc−/Gln+ condition for 24 h, followed by examining cell viability (**j**, $n = 3$ independent experiments). **k–q** H292-$KRas^{G12V}$ cells stably expressing shMT-CO2 were subjected to MTS analyses (**k**, $n = 3$ independent experiments). For tumor growth, cells ($8 \times 10^5$) were

subcutaneously inoculated into the right scruff of the female BALB/c nude mouse ($n = 7$/group). The tumor volumes on indicated days after injection were examined (**l**). Twenty days post-injection, tumors were weighed (**n**) and photos were taken (**m**). Immunohistochemistry staining analyses were performed to examine Ki67 and MT-CO2 expression (**o–q**). Scale bar, 50 μm. **r, s** The transgenic mice (Rosa26-LSL-$KRas^{G12D}$, C57BL/6) were used to assess the effects of MT-CO2 on $KRas^{G12D}$-induced lung tumor growth ($n = 5$/group), as described in Materials and Methods. Lungs were stained by H&E for histological examination (**r**). The numbers of observable nodules in the lung were presented (**s**). The samples derive from the same experiment but different gels for pERK, Ras, GAPDH, and another for ERK, MT-CO2 were processed in parallel (**a, c, e, g, i**). These experiments have been repeated three times with similar results (**a, c, e, g, i**). Data were presented as mean ± SD (**b, d, f, h, j, k**) or SEM (**l, n, p, q, s**). Comparisons were performed with one-way ANOVA with Tukey's test (**f, h, j, n, p, q**), two-way ANOVA with Tukey's test (**k, l**), and unpaired two-tailed Student's *t* test (**b, d, s**).

Next, we examined the role of MT-CO2 in $KRas^{G12V}$-mediated cell proliferation and tumor growth. As shown in Fig. 4k–q, ectopic expression of $KRas^{G12V}$ promoted H292 cell proliferation in vitro and tumor growth in vivo, which was significantly inhibited by the simultaneous silencing of MT-CO2. In the Rosa26-LSL-$KRas^{G12D}$ lung cancer mouse model, activated Ras also dramatically elevated MT-CO2 expression in $KRas^{G12D}$-driven lung tumors (Supplementary Fig. S8a, b). To further investigate the effects of MT-CO2 on tumor growth in the $KRas^{G12D}$ genetic mouse model, we designed 4 specific shRNAs for

mouse MT-CO2 (mMT-CO2). As shown in Supplementary Fig. S8c, shmMT-CO2-#2 and shmMT-CO2-#4 could effectively silence MT-CO2 expression in MEF cells. To verify that these shRNAs can be used to silence MT-CO2 in vivo, we examined the effects of lentivirus-shmMT-CO2-#2 on the MT-CO2 protein expression in the mouse lung by nasal drip. As shown in Supplementary Fig. S8d, shmMT-CO2 effectively silenced MT-CO2 expression in the mouse lung. Importantly, the silencing of MT-CO2 dramatically inhibited $KRas^{G12D}$-induced lung tumor growth (Fig. 4r, s).

Taken together, these results demonstrate that glucose starvation-mediated activation of Ras signaling stimulates the MT-CO2-GLS1 axis to promote glutaminolysis, tumor cell survival, and tumor growth.

## Activated Ras promotes *MT-CO2* transcription via facilitating TFAM expression to promote glutaminolysis and cancer cell survival in response to glucose deprivation

We next investigated the molecular basis by which activated Ras promotes MT-CO2 expression. As shown in Supplementary Fig. S9a, b, ectopic expression of KRas^G12V had little impact on mitochondrial biogenesis, as evidenced by the comparable ratio of mitochondrial DNA to nuclear DNA and citrate synthase levels. Additionally, ectopic expression of KRas^G12V had little effect on MT-CO2 mRNA stability (Supplementary Fig. S9c), indicating that the elevated MT-CO2 mRNA levels induced by activated Ras are most likely due to increased gene transcription. Since TFAM is a key mitochondrial transcription factor responsible for recruiting mitochondrial RNA polymerase to activate mitochondrial gene transcription, including *MT-CO2*[30,31], we hypothesized that TFAM may act as a critical transcription factor mediating oncogenic Ras signaling and mitochondrial MT-CO2 transcription. Indeed, activated Ras significantly upregulated TFAM mRNA and protein levels (Supplementary Fig. S9d, e). Silencing of TFAM dramatically inhibited activated Ras-induced upregulation of MT-CO2 expression and FAD levels (Supplementary Fig. S9f–h). Importantly, KRas^G12V-induced cell survival in response to glucose deprivation was significantly inhibited by the knockdown of TFAM (Supplementary Fig. S9i). Furthermore, inhibition of ERK by pharmacological inhibitor PD98059 not only effectively blocked KRas^G12V-induced upregulated of TFAM and MT-CO2 expression and FAD levels (Supplementary Fig. S9j–m), but also significantly inhibited cell viability under glucose deprivation (Supplementary Fig. S9n). Together, these results indicate that oncogenic Ras activates ERK signaling to upregulate the expression of TFAM and consequently promote *MT-CO2* transcription.

## Glucose deprivation specifically upregulates MT-CO2 expression via suppressing IGF2BP3

The human mitochondrial genome is a circular double-stranded DNA, comprising heavy (H) and light (L) strands[32]. The H-strand encodes 12 subunits of the respiratory chain complex, derived from the H-strand promoter, including MT-CO2, MT-ND1, MT-CYB, and MT-ATP6[32]. In contrast, the L-strand only encodes MT-ND6[32]. The transcription of mtDNA results in the formation of long pre-MT-mRNA, which is processed into mature MT-mRNA[32]. Therefore, the 12 subunits encoded by the H-strand should theoretically have equal mRNA levels. However, it has been reported that the levels of MT-mRNA encoded by the H-strand are variable[33,34], suggesting additional mechanisms in modulating MT-mRNA stability. Interestingly, glucose starvation led to robust elevation of MT-CO2 mRNA levels, but not other MT-mRNAs, including MT-ND1, MT-ND5, and MT-ATP6 (Fig. 1d). Given that activation of Ras had little effect on MT-CO2 mRNA stability (Supplementary Fig. S9c), these results suggest that glucose starvation specifically upregulates MT-CO2 mRNA stability in a Ras pathway-independent manner.

Accumulating evidence indicates that N6-methyladenosine (m6A) modification plays a critical role in regulating mRNA stability[35]. Yet, there are currently no reports on m6A modification of mitochondrial mRNA. Notably, as shown in Supplementary Fig. S10a, b, m6A could be found in mitochondria. RNA immunoprecipitation (RIP) analyses revealed that MT-CO2 mRNA had higher m6A modification levels than other MT-mRNAs (Fig. 5a). In addition, the analyses using the SRAMP database (http://www.cuilab.cn/sramp) also indicated that MT-CO2 has a high confidence for m6A modification (Supplementary Table 2).

m6A modification is mediated by m6A writers and erasers and is recognized by m6A readers[35]. We observed that, compared to parental H292 cells, the mRNA and protein levels of the m6A reader, IGF2BP3, were markedly reduced in glucose-starvation-resistant H292 cells (H292-V) (Fig. 5b, c). Furthermore, both the endogenous and exogenous IGF2BP3 could be found in mitochondria (Supplementary Fig. S10c–f). These results suggest that IGF2BP3 could specifically regulate MT-CO2 expression. Indeed, silencing IGF2BP3 specifically upregulated MT-CO2 mRNA and protein levels in H1299 cells (Fig. 5d, e). Conversely, ectopic expression of IGF2BP3 specifically suppressed MT-CO2 mRNA and protein levels (Fig. 5f, g).

We then examined the effects of IGF2BP3 on MT-CO2 mRNA stability. As shown in Fig. 5h, the knockdown of IGF2BP3 significantly upregulated MT-CO2 mRNA stability. In contrast, ectopic expression of IGF2BP3 markedly reduced MT-CO2 mRNA stability (Fig. 5i). Moreover, RIP analyses showed that IGF2BP3 could bind to MT-CO2 mRNA, but not to other MT-mRNAs, such as MT-CYB, MT-ND1, MT-CO1, MT-ND3, or MT-ND4L (Fig. 5j). Importantly, restoration of IGF2BP3 in H292-V cells markedly reduced MT-CO2 expression to levels similar to those in parental H292 cells (Fig. 5k).

Together, these results indicate that IGF2BP3 specifically inhibits MT-CO2 mRNA stability. Glucose starvation upregulates MT-CO2 expression via downregulation of IGF2BP3.

## MT-CO2 expression is linked to activated Ras and the levels of c-JUN and GLS1 in human lung cancer samples

Next, we investigated the clinical relevance of this study. We first examined MT-CO2 and GLS1 protein expression in human lung cancer samples. As shown in Fig. 6a–c, both MT-CO2 and GLS1 protein expression were significantly upregulated in lung cancer samples compared to adjacent tissues. Importantly, the protein expression of MT-CO2 and GLS1 exhibited a significant and positive correlation in clinical lung cancer samples (r = 0.69, *p* < 0.0001; Fig. 6d). In addition, lung cancer samples with *KRAS* mutations showed higher mRNA levels of MT-CO2, c-JUN, or GLS1 (Fig. 6e–g). Notably, a positive correlation was observed regarding mRNA expression between MT-CO2 and GLS1 (r = 0.66, *p* = 0.0005), or between MT-CO2 and c-JUN (r = 0.55, *p* < 0.0001) (Fig. 6h, i). Furthermore, lung cancer patients with higher mRNA levels of MT-CO2 exhibited lower overall survival (OS) (Fig. 6j). Consistently, Kaplan–Meier database analyses showed that lung cancer patients with higher mRNA levels of GLS1 had lower overall survival (OS) (Fig. 6k).

## Berberine inhibits the MT-CO2-GLS1 axis to suppress oncogenic Ras-induced glutaminolysis and lung tumor growth

Our abovementioned results show that MT-CO2 plays a critical role in oncogenic Ras-induced glutaminolysis and tumor growth. We thus rationalized that cancer-associated elevation of MT-CO2 may be a putative therapeutic target for Ras-driven cancers. To search for potential small chemical MT-CO2 inhibitors, we examined the effects of several known mitochondrial respiratory chain inhibitors (metformin[36], oligomycin, rotenone, and berberine[37]), ROS inducers (elesclomol[38], tolperisone[39], and piperlongumine[40]) or glycolysis inhibitor (2-Deoxy-D-glucose, 2-DG) on MT-CO2 expression. As shown in Fig. 7a and Supplementary Fig. S11a, berberine, a clinically approved drug used to treat diarrhea and diabetes[41], remarkably inhibited MT-CO2 expression in Ras-activated A549 and H1299 cells, concomitant with reduced GLS1 expression in a dose-dependent manner. Notably, berberine inhibited MT-CO2 mRNA expression without affecting mitochondrial mass (Supplementary Fig. S11b, c), suggesting that berberine inhibits *MT-CO2* transcription. In keeping with our aforementioned observation that TFAM upregulates *MT-CO2* gene transcription (Supplementary Fig. S9f, g), berberine significantly inhibited the expression of TFAM, accompanied by the suppression of MT-CO2, and GLS1 expression (Supplementary Fig. S11d). Importantly, ectopic expression of TFAM effectively rescued berberine-induced downregulation of MT-CO2 expression (Supplementary Fig. S11e, f).

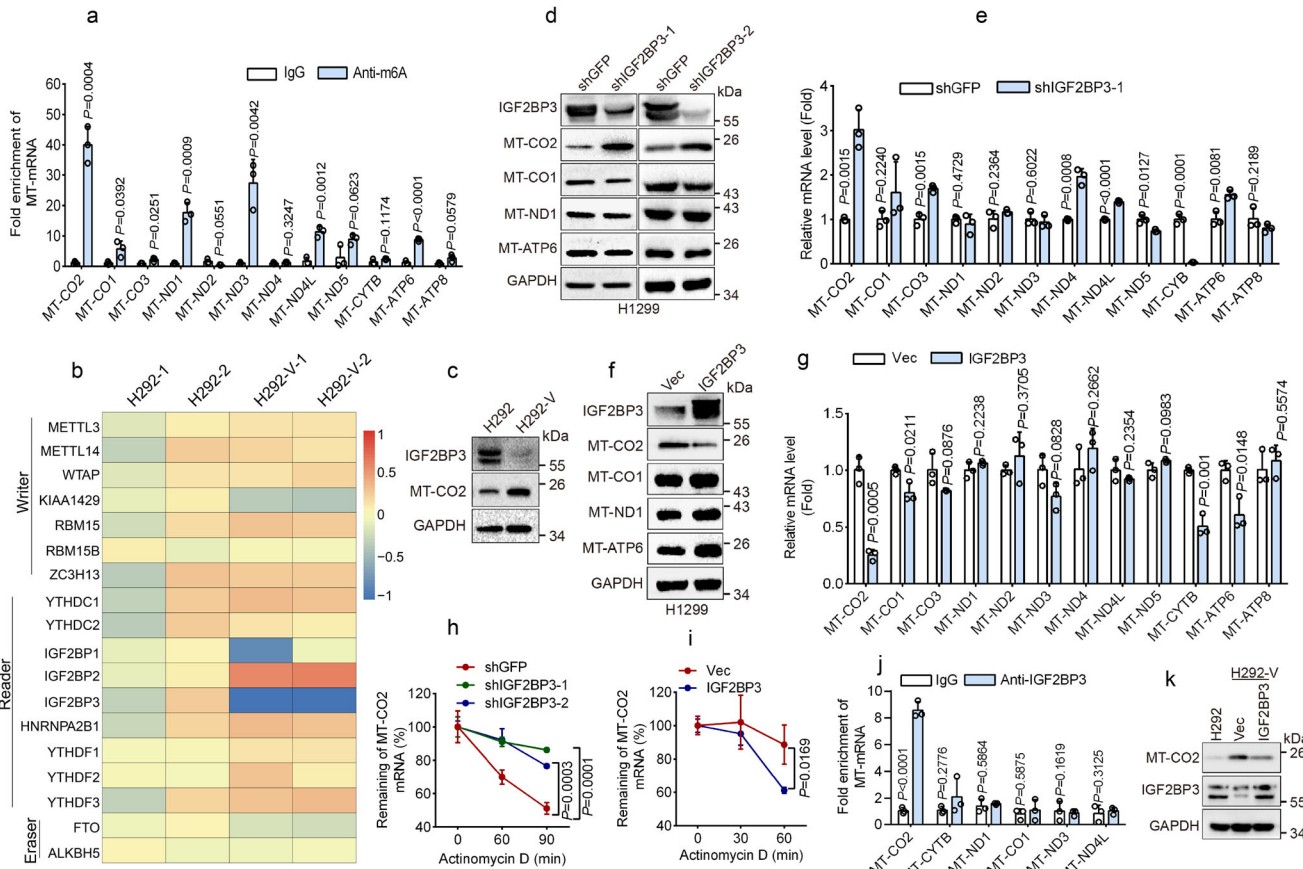

**Fig. 5 | Glucose starvation specifically facilitates MT-CO2 expression via downregulation of IGF2BP3. a** m6A RIP-qPCR was performed to analyze the levels of m6A-modified mitochondrial genes in H1299 cells (*n* = 3 independent experiments). **b, c** H292 or glucose-starvation-resistant H292 (H292-V) cells were subjected to RNA-seq analyses (b, *n* = 2 biologically independent samples per experiment) or were subjected to western blot analyses (c). **d, e** H1299 cells expressing shGFP, shIGF2BP3-1, or IGF2BP3-2 were subjected to western blot (**d**) or qPCR (**e**, *n* = 3 independent experiments) analyses. **f, g** H1299 cells expressing IGF2BP3 or vector control (Vec) were subjected to western blot (**f**) or qPCR (**g**, *n* = 3 independent experiments) analyses. The samples derive from the same experiment but different gels for IGF2BP3, MT-CO2, GAPDH, another for MT-ND1, and another

for MT-CO1, MT-ATP6 were processed in parallel (**d, f**). **h, i** H1299 cells expressing shGFP, shIGF2BP3-1, or IGF2BP3-2 (**h**) or H1299 cells expressing IGF2BP3 or Vec (**i**) were treated with actinomycin D (10 μM) for an indicated time point. Cells were subjected to qPCR analyses. The half-life of the MT-CO2 mRNA levels was shown (*n* = 3 independent experiments). **j** IGF2BP3 RIP-qPCR was performed to analyze the interaction between IGF2BP3 and mitochondrial mRNAs in H1299 cells (*n* = 3 independent experiments). **k** H292-V cells expressing IGF2BP3 or Vec were subjected to western blot analyses. These experiments have been repeated three times with similar results (**c, d, f, k**). Data were presented as mean ± SD (**a, e, g, h–j**). Comparisons were performed with two-way ANOVA with Tukey's (**h**) or Bonferroni's (**i**) test and unpaired two-tailed Student's *t* test (**a, e, g, j**).

---

Together, these results indicate that berberine inhibits *MT-CO2* transcription via suppression of TFAM.

We further investigated the effects of berberine on oncogenic Ras-induced glutaminolysis and cell proliferation. As shown in Supplementary Fig. S11g, h, berberine dramatically suppressed A549 or H1299 cell growth, which was markedly rescued by a supplement of α-KG, suggesting that berberine inhibits glutaminolysis. In supporting this notion, berberine-mediated inhibition of cell viability under the Glc-/Gln+ (DMEM containing 2 mM glutamine in the absence of glucose) condition was largely rescued by a supplement of α-KG (Fig. 7b). Furthermore, berberine strongly inhibited KRas[G12V]-induced upregulated MT-CO2 and GLS1 expression, resulting in suppression of cell proliferation and cell viability, both of which could be markedly rescued by supplement of α-KG (Fig. 7c, d and Supplementary Fig. S11i). Notably, the oncogenic Ras-induced upregulation of MT-CO2 and GLS1 expression as well as lung tumor growth were significantly inhibited by berberine treatment in vivo (Fig. 7e–h and Supplementary Fig. S11j, k). Moreover, berberine significantly inhibited A549-derived xenograft tumor growth, concomitant with a marked reduction in MT-CO2 and GLS1 expression (Fig. 7i–l and Supplementary Fig. S11l, m). We further investigated whether berberine can also inhibit the growth of Ras

mutant tumors in mice with a healthy immune system. As shown in Fig. 7m–o and Supplementary Fig. S11n, berberine not only significantly suppressed tumor growth derived from mouse LLC cells (Lewis lung carcinoma cell line harboring a *KRAS[G12C]* allele) but also prolonged the overall survival of the C57BL/6 J mice. Importantly, a low dose of berberine treatment significantly suppressed the proliferation of *RAS* mutant tumor cells, while having little effect on the growth of tumor cells bearing wild-type *RAS* alleles (Supplementary Fig. S12).

Together, these results indicate that berberine is an effective MT-CO2 inhibitor and highlight that MT-CO2 is a valuable therapeutic target for oncogenic Ras-driven cancers.

## Discussion

Glucose deprivation is often associated with tumor development, forcing tumor cells to seek alternative energy sources for survival and growth. This study demonstrates that MT-CO2 promotes metabolic reprogramming by switching the energy source from glucose to glutamine in response to glucose deprivation. We show that glucose deprivation activates Ras signaling while suppressing IGF2BP3, leading to the upregulation of MT-CO2 expression, which in turn facilitates glutaminolysis. We provide compelling evidence that MT-CO2 is not

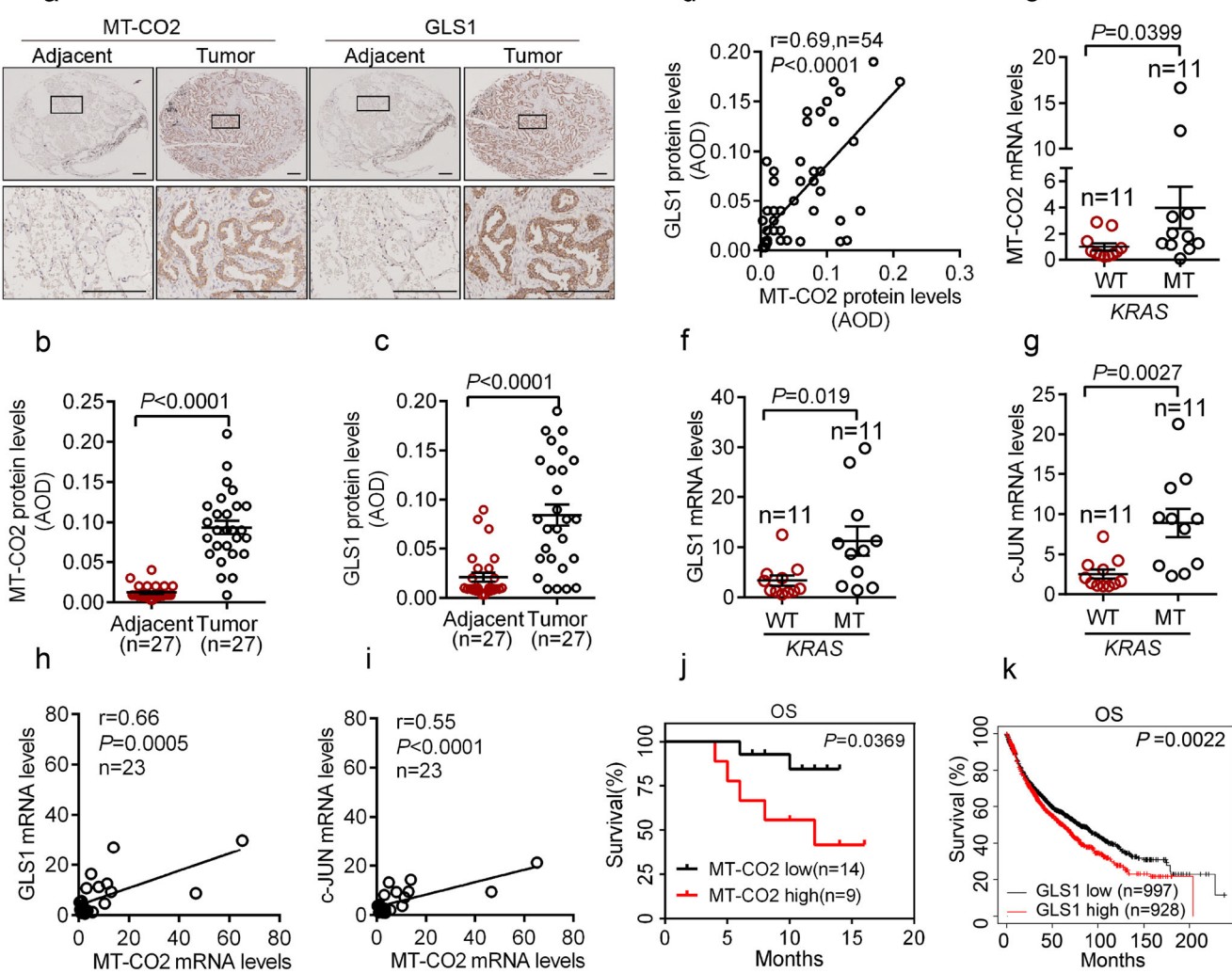

**Fig. 6 | Elevated MT-CO2 expression is linked to oncogenic Ras and is positively correlated with the expression of c-JUN and GLS1, and with poor clinical outcomes in human lung cancer. a–d** Human lung cancer tissue microarrays consisting of 27 pairs of tumors and adjacent tissues were subjected to immuno-histochemistry analyses for MT-CO2 and GLS1 expression (**a**) with quantitative analyses using average optical density (AOD) (**b**, **c**). The Pearson correlation between MT-CO2 and GLS1 in protein levels was analyzed (**d**). Scale bar, 200 μm. **e–g** Human lung tumor specimens harboring wild-type *KRAS* alleles (*KRAS*^WT, *n* = 11) or a *KRAS* mutant allele (*KRAS*^MT, *n* = 11) were qPCR analyzed for MT-CO2 (**e**),

GLS1(**f**), and c-JUN (**g**) mRNA levels. **h–j** Human lung tumor samples (*n* = 23) were qPCR analyzed for MT-CO2, GLS1, and c-JUN mRNA levels. The Pearson correlation between MT-CO2 and GLS1 (**h**) or c-JUN (**i**) in mRNA levels was analyzed. The correlation between MT-CO2 mRNA levels and the overall survival (OS) in lung cancer patients was analyzed (**j**). **k** The correlation between GLS1 mRNA levels and overall survival (OS) in lung cancer patients was analyzed using the Kaplan–Meier Plotter database. Data were presented as mean ± SEM (**b**, **c**, **e**, **f**, **g**). Statistical analyses were performed with unpaired two-tailed Student's *t* test (**b**, **c**, **e**, **f**, **g**), Log-rank (Mantel–Cox) test (**j**, **k**), and two-tailed Pearson correlation analysis (**d**, **h**, **i**).

only essential for tumor cell survival upon glucose deprivation but is also indispensable for KRas-driven glutaminolysis and tumor growth. Mechanistically, we show that increased MT-CO2 upregulates flavin adenosine dinucleotide (FAD) levels to activate lysine-specific histone demethylase 1 (LSD1), which epigenetically upregulates *JUN* transcription via suppressing H3K9me2 levels, consequently resulting in increased GLS1 expression and glutaminolysis (Fig. 7p).

It is well-established that glucose is a major energy source for tumor cell survival and growth. However, it remains a central issue how tumor cells sense glucose shortage and adapt to alternative energy sources. Tumor cells can facilitate fatty acid oxidation or glycogen catabolism to maintain survival in response to glucose deprivation[15,17]. These processes are regulated by highly conserved energy sensors and signaling hubs, such as AMP-activated protein kinase (AMPK) and mTORC1, both critical for cell survival under glucose deprivation[42]. Glucose starvation can activate AMPK to facilitate lipid droplet lipo-lysis and NADPH homeostasis, thereby promoting tumor cell

survival[43,44]. Inhibition of mTORC1 can suppress oxidative stress by maintaining NADPH homeostasis, aiding tumor cell survival under glucose deprivation[42]. A recent study reports that glucose shortage leads to AMPK-mediated phosphorylation of PDZD8, a scaffold protein binding to and activating GLS1 enzymatic activity to promote gluta-minolysis ahead of utilization of fatty acid[45]. Here, we demonstrate that glucose starvation enhances TFAM-mediated MT-CO2 expression by activating Ras signaling, which promotes GLS1 expression, glutami-nolysis, and tumor cell survival, revealing that the Ras-MT-CO2-GLS1 axis as a pathway that triggers a shift in carbon source from glucose to glutamine. These observations point to the pleiotropic effects in the cellular response to glucose deprivation. It is possible that the Ras-MT-CO2 axis promotes GLS1 expression, whereas AMPK signaling enhan-ces GLS1 activity upon glucose shortage, both promoting glutamino-lysis in supporting tumor cell viability and growth.

These findings position MT-CO2 as a crucial regulator in orches-trating metabolic reprogramming to sustain tumor cell survival under

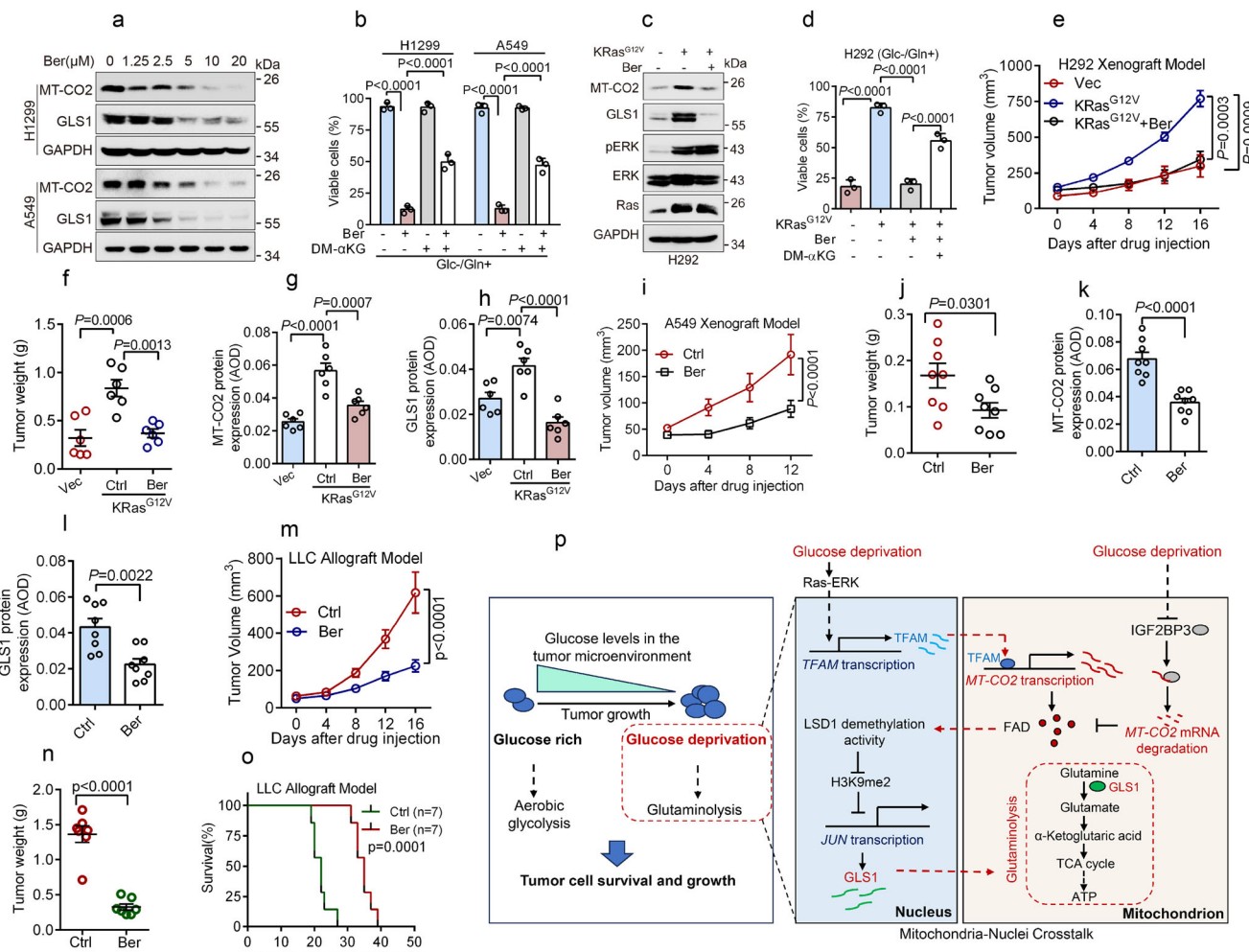

**Fig. 7 | Berberine inhibits the MT-CO2-GLS1 axis to suppress oncogenic Ras-induced glutaminolysis and tumor growth. a, b** Indicated cells were treated with an indicated concentration of berberine for 24 h prior to western blot analyses (**a**) or were treated with berberine (5 μM, and thereafter) in the presence of 500 μM dimethyl α-Ketoglutaric acid (DM-αKG) upon Glc−/Gln+ condition for 24 h, followed by examining cell viability (**b**, n = 3 independent experiments). **c, d** H292-KRas^G12V cells were treated with berberine for 24 h prior to western blot analyses (**c**) or were treated with berberine in the presence of 500 μM DM-αKG upon Glc−/Gln+ condition for 24 h, followed by examining cell viability (**d**, *n* = 3 independent experiments). The samples derive from the same experiment but different gels for MT-CO2, GLS1, GAPDH, another for pERK, Ras, and another for ERK were processed in parallel (**c**). **e–l** H292-KRas^G12V (8 ×10⁵, **e–h**) or A549 (1 ×10⁶, **i–l**) cells were subcutaneously inoculated into the right scruff of the female BALB/c nude mouse (*n* = 6/group, H292; *n* = 8/group, A549). On day 14 (H292) or 22 (A549) after inoculation, mice were given berberine (40 mg/kg) by oral gavage daily. The tumor

volumes on indicated days after gavage were examined (**e, i**). On day 16 (H292) or 12 (A549) after gavage, tumors were weighed (**f, j**). Immunohistochemistry staining was performed to examine MT-CO2 and GLS1 expression (**g, h, k, l**). **m–o** Lewis lung carcinoma (LLC, 5 ×10⁵) cells were subcutaneously inoculated into the right scruff of each C57BL/6 mice (*n* = 7/group). On day 7 after inoculation, mice were given berberine (40 mg/kg) by oral gavage daily. The tumors volume on indicated days after gavage were examined (**m**). On day 16 after gavage, tumors were weighed (**n**). The mice's overall survival (Tumor volume >1500 mm³ is defined as ethical death) was examined (*n* = 7/group) (**o**). **p** A model depicting the role of MT-CO2 in tumor cell survival upon glucose deprivation. These experiments have been repeated three times with similar results (**a, c**). Data were presented as mean ± SD (**b, d**) or SEM (**e–n**). Comparisons were performed with one-way ANOVA with Tukey's test (**b, d, f–h**), two-way ANOVA with Tukey's (**e**) or Bonferroni's (**i, m**) test, unpaired two-tailed Student's *t* test (**j–l, n**), and Log-rank (Mantel–Cox) test (**o**).

glucose deprivation. Notably, MT-CO2 is not only essential in promoting glutaminolysis for cancer cell survival but also is indispensable for tumor cell growth in vitro and in vivo. Given that glutamine addiction is a key feature in metabolic reprogramming for tumor growth[46], it is plausible that MT-CO2 also plays a crucial role in glutamine addiction through the stimulation of glutaminolysis.

Our results show that glucose starvation activates Ras signaling to upregulate the transcription of *MT-CO2*. It is known that MT-CO2, along with 11 other MT-mRNAs, is encoded by the heavy (H) strand of the mitochondrial genome through a single H-strand promoter[32] so that the H-strand promoter would drive the transcription to yield an equal amount of pre-mRNAs for each gene. Then, a critical issue is how glucose starvation specifically upregulates MT-CO2 expression. Our

results show that glucose starvation reduces the expression of IGF2BP3, a well-known N6-methyladenosine (m6A) modification reader. Importantly, IGF2BP3 can specifically bind to and destabilize MT-CO2 mRNA, revealing that m6A modification can also applied to mitochondrial DNA-encoded mRNA. Therefore, this study reveals that glucose deprivation-mediated robust upregulation of MT-CO2 expression is achieved by Ras-mediated transcription upregulation and IGF2BP3-mediated modulation of MT-CO2 mRNA stability.

Abundant evidence demonstrates that glutaminolysis is crucial in promoting tumor growth[47]. Genetic or pharmacological inhibition of GLS1 or SLC1A5 significantly suppresses tumor growth in vivo[48–50]. Despite previous reports that the GLS1 inhibitor CB-839 failed to suppress KRas-driven pancreatic ductal adenocarcinoma growth[51] and

showed no advantage over standard immunotherapy in stage IV non-small cell lung cancer[52], a phase 1b/2 trial found that CB-839 combined with azacytidine is a safe and effective treatment for myelodysplastic syndrome[53]. In addition, a phase II trial showed that its combination with everolimus significantly improved progression-free survival in renal cell carcinoma patients[54]. These developments highlight the therapeutical potential of targeting glutaminolysis in cancer treatment.

It is well-known that complexes I-IV of the mitochondrial respiratory chain are responsible for oxidative phosphorylation. Our findings show that inhibition of respiratory chain complexes II-IV by either genetic maneuvers or by small molecule inhibitors significantly suppresses c-JUN/GLS1 expression via inactivation of the FAD-LSD1-H3K9me2 axis. In contrast, inhibition of respiratory chain complex I has little effect on FAD levels and does not affect c-JUN/GLS1 expression. Together, these results uncover a critical function for respiratory chain complexes II-IV in regulating glutaminolysis and provide strong evidence for a mitochondrial retrograde signaling pathway through MT-CO2-mediated epigenetic regulation of GLS1.

Given glutaminolysis is essential in supporting various biological processes, it is plausible that defective MT-CO2 may be involved in human diseases. Indeed, It has been reported that supplementing glutamine can reduce the risk of stroke and coronary artery disease (CAD)[55], suggesting that glutaminolysis is involved in the development of these diseases. Notably, a clinical study reported that mitochondrial *MT-CO2* gene 8231C>A (L216M) missense mutation is significantly associated with CAD ($p = 0.007$)[56], the molecular basis of which is not yet known. It might be possible that MT-CO2$^{L216M}$ is defective in the regulation of GLS1 expression, leading to the inhibition of glutaminolysis.

Targeting Ras remains a vast challenge for cancer therapy, despite the recent clinical approval of AMG-510[57], a specific small molecular inhibitor of KRas$^{G12C}$, which is found in approximately 13% of lung adenocarcinomas and 1–3% of other solid tumors[57]. A recent study has reported that tumor cells rapidly acquire resistance to small molecule inhibitors of KRas$^{G12C}$[58]. Therefore, it is urgent to develop therapeutic strategies for Ras-activated cancers. Here, we show that either knockdown of MT-CO2 or berberine significantly suppresses Ras-induced tumor growth in vivo, highlighting MT-CO2 as a putative therapeutic target for treating Ras-activated cancers.

## Methods
### Ethics statement and mouse models
All animal care and animal experiments in this study were performed in accordance with China's National Legislation and the institutional ethical guidelines and were approved by the institutional review board of Sichuan University (Approval Number: SCU23030314). Mice were fed a standard chow diet and were maintained in individual cages at a room temperature of $22 \pm 2\,°C$ on a 12: 12 light–dark cycle (lights on at 09:00 h).

For in vivo tumor growth, six-week-old female BALB/c nude mice (BALB/cNj-Foxn1nu/Gpt) were purchased from GemPharmatech (Chengdu, China). Indicated cells were subcutaneously inoculated into the right scruff of each nude mouse. Mice were monitored for tumor size and sacrificed on the indicated day. Tumor weight, volume, and photos were taken. The xenograft tumor samples were subjected to IHC staining analyses. Tumor size was measured with a caliper and tumor volume was calculated by width$^2 \times$ length $\times 1/2$. The maximal tumor size permitted by the IACUC of Sichuan University is 20 mm at the largest diameter in mice. The maximal tumor size in our animal experiments did not exceed the permitted maximal tumor size.

For transgenic mice, eight-week-old LSL-*Kras*$^{G12D}$ knock-in mice (C57BL/6 background, provided by Dr. Chong Chen of Sichuan University) were infected with $2 \times 10^7$ PFU adenovirus-based Cre by nasal drip. 28 days post-infection, mice were infected with $2 \times 10^7$ PFU

lentivirus-based shGFP or shMT-CO2 by nasal drip. On day 56 after the lentivirus infection, lungs were dissected, fixed in 4% polyformaldehyde, embedded in paraffin, and sectioned onto microscope slides for hematoxylin and eosin (H&E) staining before histological analysis. The number of nodules in the lungs per mouse was counted.

### Cell culture and drug treatment
H1975 (CRL-5908), H292 (CRL-1848), and LLC (Lewis lung carcinoma, CRL-1642) were obtained from ATCC (Manassas, VA, USA). H1299 (CL-0165) were obtained from Procell Life Science&Technology (Wuhan, China). HEK-293FT (R70007) was obtained from Thermo Fisher Scientific (Waltham, MA, USA). A549 (BNCC337696) was obtained from the BeNa Culture Collection (Beijing, China). MEF (mouse embryonic fibroblast) cells were isolated from day 13.5 mouse embryos. All cell lines used in this study were routinely tested to be negative for mycoplasma contamination and were kept at low passages to maintain their identity and were authenticated by morphology check and growth curve analysis.

A549, H1299, HEK 293FT, and MEF cells were cultured in DMEM medium (GIBCO, Rockville, MD, USA) supplemented with 10% fetal bovine serum (FBS; Hyclone, Logan, UT, USA). Human lung mucoepidermoid pulmonary carcinoma H292 and human NSCLC H1975 cells were cultured in RPMI-1640 medium (GIBCO) supplemented with 10% FBS (Hyclone). All cells were grown in a medium supplemented with 2 mM glutamine, 100 units/mL penicillin (GIBCO), and 100 μg/mL streptomycin (GIBCO) with or without 4.5 mg/mL glucose as indicated. Cells were maintained in a humidified 37 °C incubator under a 5% $CO_2$ atmosphere. Cells at 75–85% confluence were treated with an indicated chemical compound(s). PD98059 (S1177), oligomycin (S1478), rotenone (S2348), elesclomol (S1052), tolperisone (S4200), 2-Deoxyglucose (2-DG, S4701), ATP (S1985), and nicotinamide mononucleotide (NMN, S5259) were purchased from Selleck (Houston, TX, USA). Berberine (HY-18258), carboxin (HY-B2064), antimycin A (HY-107406), flavin adenine dinucleotide (FAD+, HY-B1654A), riboflavin (HY-B0456), CB-839 (HY-12248), CP-91149 (HY-13525), etomoxir (HY-50202), and GSK-LSD1 (HY-100546A) were purchased from MCE (Monmouth Junction, NJ, USA). Metformin (PHR1084), dimethyl α-ketoglutarate (DM-αKG, 349631), and piperlongumine (SML0221) were purchased from Sigma (St. Louis, MO, USA). Sodium acetate (NaAc, ST351) was purchased from Beyotime (Beijing, China).

### Plasmid transfection, lentiviral infection, and RNA interference
Cells at 70–80% confluence were transfected using Lipofectamine 2000 transfection reagent (Invitrogen, USA). Expression plasmids were used in this study including KRas$^{G12V}$, HRas$^{G12V}$, TFAM, GLS1, c-JUN, or IGF2BP3. Recombinant lentiviruses were amplified by transfecting HEK-293FT cells with psPAX2 and pMD2.G packaging plasmids and lentiviral expression plasmid using Lipofectamine 2000. Viruses were collected at 60 h after transfection. Cells at 50% confluence were infected with a recombinant lentivirus in the presence of 10 μg/mL polybrene, followed by 12 h of incubation at 37 °C with 5% $CO_2$. Lentiviral-based shRNAs specific for green fluorescent protein (GFP), human MT-CO2, TFAM, MT-ND1, MT-CYB, MT-ATP6, IGF2BP3, or mouse MT-CO2 were constructed into a pLKO.1-puromycin lentiviral vector. The shRNA sequences were listed in Supplementary Table 3.

### Western blot, immunohistochemistry, and immuno-fluorescence staining analyses
For Western blot, cells were collected, washed with cold PBS, and resuspended in EBC$_{250}$ lysis buffer (250 mM NaCl, 50 mM Tris pH 8.0, 0.5% Nonidet P-40, 50 mM NaF, 1 mM phenylmethylsulfonyl fluoride, 2 μg/mL aprotinin, and 2 μg/mL leupeptin). Equal amounts of total protein were loaded, separated by SDS-PAGE, transferred to PVDF membranes (Millipore, Darmstadt, Germany), and hybridized to an appropriate primary antibody and horseradish peroxidase (HRP)-

conjugated secondary antibody for subsequent detection by enhanced chemiluminescence (Bio-Rad). Western blot images were analyzed using Image Lab Software 5.0. Antibodies for Ras (3965, 1:1000), pERK (Thr202/Tyr204, 9101, 1:1000), ERK (9102, 1:1000), EZH2 (5246, 1:1000), c-JUN (9165, 1:1000), HA-Tag (5017, 1:1000), H3K27me3 (9733, 1:1000), H3K27ac (8173, 1:1000), pAMPK (Thr172, 2535, 1:1000), and AMPK (2532, 1:1000) were purchased from Cell Signaling Technology (CST, Danvers, MA, USA). Antibodies for GLS1 (CY5719, 1:1000), MT-CO2 (human, CY5717, 1:1000), MT-ND1 (CY8176, 1:1000), H3K4me2 (CY5123, 1:1000), H3K9me2 (CY5761, 1:1000), LSD1 (CY5517, 1:1000), p21 (CY5543, 1:1000), and GAPDH (AB0037, 1:5000) were purchased from Abways (Shanghai, China). Antibodies for MT-CO2 (mouse, ab198286, 1:1000) and TFAM (ab131607, 1:2000) were purchased from Abcam (Cambridge, UK). Antibody for MT-CO1 (DF8920, 1:1000) was purchased from Affinity (Jiangsu, China). Antibodies for MT-ATP6 (A8193, 1:1000), Citrate synthase (A23371, 1:1000), and SLC1A5 (A6981, 1:1000) were purchased from ABclonal (Wuhan, China). Antibody for p53 (SC-126, 1:300) was purchased from Santa Cruz (CA, USA). Antibody for IGF2BP3 (14642-1-AP, 1:1000) was purchased from Proteintech (Wuhan, China).

For immunohistochemistry (IHC) analyses, human lung adenocarcinoma tissue microarrays (TMA) (HLugA060PG02) were obtained from Outdo Biotech (Shanghai, China). Paraffin-embedded tumors or lung samples were sliced into 5 μm thickness. Tissue sections were rehydrated through a decreasing ethanol gradient, and treated by boiling in citrate buffer (pH 6.0) or Tris-EDTA (pH 9.0) for antigen retrieval. Endogenous peroxidases were blocked using 0.3% $H_2O_2$. After blocking with 5% BSA, the sections were incubated with primary antibody and followed by HRP-conjugated secondary antibody. The sections were subsequently stained with a DAB Detection Kit (ZLI-9018, ZSGB-BIO, China). Antibodies used in IHC for GLS1 (CY5719, 1:200) and MT-CO2 (human, CY5717, 1:200) were purchased from Abways (Shanghai, China). Antibody for human Ki67 (9449, 1:800) was purchased from CST (Danvers, MA, USA). The antibody for mouse Ki67 (ab15580, 1:200) was purchased from Abcam (Cambridge, UK). Slides were scanned through NanoZoomer (Hamamatsu, Japan). Scanned images were then subjected to integrated optical density (IOD) measurements using Image-Pro Plus 6.0 to calculate the average optical density (AOD) using the formula: AOD = IOD/Area[59].

For Immunofluorescence staining, cells grown on coverslips were fixed with 4% polyformaldehyde in PBS, permeabilized with 0.1% Triton X-100 in PBS, blocked with 4% bovine serum albumin in PBS, hybridized to an appropriate primary antibody anti-m6A (m6A: 1:500, 68055-1-Ig, Proteintech; MT-CO2: 1: 200, CY5717, Abways; IGF2BP3: 1:200, 14642-1-AP, Proteintech; Flag-Tag: 1:200, 8146, CST; Hsp60: 1:200, 66041-1-Ig, Proteintech), followed by incubation with a second antibody (Goat anti-Mouse Alexa Fluor 488, A-11029 or Goat anti-Rabbit Alexa Fluor 514, A-31558, ThermoFisher). The cells were counterstained with ProLong® Gold Antifade Reagent with DAPI (#8961, CST) prior to visualization and photographed using a Leica TCS SP5II confocal laser scanning microscope. LAS X (V3.3.0) was used to analyze fluorescent images. To determine the IGF2BP3 or m6A co-localization with Hsp60 or MT-CO2, the free software Image J. Fiji coupled with the Coloc 2 plugin and Pearson's correlation coefficient were used to calculate double fluorescence correlation coefficients[60], and co-localized fluorescence quantifications were presented by scatter diagrams.

### Quantitative PCR (qPCR) and measurement of the ratio of mitochondrial DNA/nuclear DNA

Total RNA was extracted from cells using an RNA extraction kit (QIA-GEN, Germany) followed by reverse transcription using ReverTra Ace qPCR RT Master Mix (FSQ-201, TOYOBO, Japan). qPCR was carried out for MT-CO2, TFAM, GLS1, GLUD1, c-JUN, and GAPDH (The primer

sequences used in the reactions were listed in Supplementary Table 3). The qPCR reactions were performed in CFX-960 Real-time PCR System (Bio-Rad) and using Bio-Rad SoFast Eva-Green Supermix Kit (Bio-Rad) according to the manufacturer's instructions. Relative quantitation values were calculated using the ΔΔCt method. For measurement of the ratio of mitochondrial DNA/nuclear DNA (mtDNA/nDNA), nuclear DNA and mitochondrial DNA were extracted from cells using the TIANamp Genomic DNA Kit (TIANGEN BIOTECH, China). qPCR was carried out for CYTB and H2B (The primer sequences used in the reactions were listed in Supplementary Table 3). The ratio of mtDNA/nDNA = CYTB copies/H2B copies.

### Chromatin immunoprecipitation (ChIP) assay

ChIP assays were performed using ChIP-IT Kit (Active Motif, USA) using antibodies specific for H3K9me2 (CY5761, Abways), H3K4me2 (CY5123, Abways) or normal rabbit IgG (2729, Cell Signaling Technology). To examine the strength of H3K9me2 or H3K4me2 for binding to *JUN* gene promoter elements, ChIP samples were subjected to qPCR using primers as listed in Supplementary Table 3. The value of each ChIP sample was normalized to its corresponding input.

### RNA immunoprecipitation (RIP) assay

The RIP assay was carried out using a BeyoRIP™ RIP Assay Kit (P1801S, Beyotime, China), according to the manufacturer's instructions. Briefly, $1 \times 10^7$ cells were lysed in complete RIP lysis buffer containing protein A/G Agarose conjugated with anti-m6A (68055-1-Ig, Proteintech) or anti-IGF2BP3 (14642-1-AP, Proteintech) at 4 °C for 4 h. RNA was then purified using TRIzol™ Reagent (15596026, Invitrogen) and used for qPCR analysis.

### RNA sequencing and whole exon sequencing

For RNA sequencing analyses, total RNA was extracted using TRIzol reagent (Thermo Fisher Scientific, USA) according to the manufacturer's protocol. RNA purity and quantification were evaluated using the NanoDrop 2000 spectrophotometer (Thermo Scientific, USA). RNA integrity was assessed using the Agilent 2100 Bioanalyzer (Agilent Technologies, Santa Clara, CA, USA). Then the libraries were constructed using TruSeq Stranded mRNA LT Sample Prep Kit (Illumina, San Diego, CA, USA) according to the manufacturer's instructions. The libraries were sequenced on an Illumina HiSeq X Ten platform and 150 bp paired-end reads were generated and analyzed (OE Biotech Co., Ltd, Shanghai, China). Differential expression analysis was performed using the DESeq (2012) R package. P value < 0.05 and foldchange >2 or foldchange <0.5 was set as the threshold for significantly differential expression. KEGG pathway enrichment analysis of differentially expressed genes was performed respectively using R based on the hypergeometric distribution.

Whole exon sequencing and data analysis were conducted by Seqhealth Technology Co., LTD (Wuhan, China). DNA was extracted from cells using a commercial kit (TIANGEN Biotech, Beijing, Chin). Genomic DNA integrity was checked by 0.5% agarose gel electrophoresis. Qualified DNA was finally quantified by Qubit3.0 with QubitTM RNA Broad Range Assay kit (Life Technologies, Q10210). Libraries of DNA were prepared using 50-200 ng DNA, captured by xGenTM Exome Hyb Panel v2 (Integrated DNA Technologies) following the manufacturer's instructions. PCR products corresponding to 200–500 bps were enriched, quantified, and finally sequenced on a Novaseq 6000 sequencer (Illumina) with PE150 mode. The raw FastQ files were first filtered by fastp and reads were mapped to the hg19 reference genome. Both single nucleotide polymorphism (SNP) and insertion/deletion (INDEL) were called, filtered, and annotated using GATK HaplotypeCaller, GATK VariantFiltration, and ANNOVAR, respectively. The sequencing depth of the chromosomes and the distribution of the variants were visualized using Circos.

### Detection of targeted metabolites

$1 \times 10^7$ cells were frozen in liquid nitrogen for 10 min. Extraction and analysis of metabolites were carried out by Metware Biotechnology Co. Ltd. (Wuhan, China) based on the AB Sciex QTRAP 6500 LC-MS/MS platform. Briefly, the sample was thawed on ice, then mixed with 500 μL of 80% methanol/water (precooled at −20 °C), and then frozen in liquid nitrogen for 5 min, removed on ice for 5 min, after that, the sample was vortexed for 2 min. The previous step was repeated 3 times. The sample was centrifuged at $13,000 \times g$ for 10 min at 4 °C. Take 300 μL of supernatant into a fresh centrifuge tube and place the supernatant in a −20 °C refrigerator for 30 min. Then the supernatant was centrifuged at $13,000 \times g$ for 10 min at 4 °C. After centrifugation, 200 μL of supernatant were analyzed using an LC-ESI-MS/MS system (UPLC, ExionLC AD, https://sciex.com.cn/; MS, QTRAP® 6500+ System, https://sciex.com/). All metabolites were detected by MetWare (http://www.metware.cn/) based on the AB Sciex QTRAP 6500 LC-MS/MS platform. Significantly regulated metabolites between groups were determined by VIP and absolute Log2FC (foldchange).

### $^{13}C_5$-glutamine metabolic flux analysis

Cells were grown in DMEM containing 2 mM $^{13}C_5$-glutamine for 8 h. After incubation, the cells were ground in liquid nitrogen and resuspended in 1 mL of cold (−40 °C) 50% aqueous methanol. The suspension was placed on dry ice for 30 min, then thawed on ice. Subsequently, 0.4 mL of chloroform was added, and the mixture was vortexed for 30 s. The samples were centrifuged at $14,000 \times g$ for 15 min at 4 °C, and the resulting supernatant was transferred to a fresh 1.5 mL tube and stored at −80 °C until further analysis. Water-soluble metabolites were analyzed using a Q Exactive PLUS hybrid quadrupole-orbitrap mass spectrometer (Thermo Scientific) coupled to hydrophilic interaction chromatography (HILIC). Separation was performed on an XBridge BEH Amide column (150 mm × 2.1 mm, 2.5 μM particle size, Waters, Milford, MA) using a solvent gradient of A (95% $H_2O$:5% acetonitrile with 20 mM ammonium acetate and 20 mM ammonium hydroxide, pH 9.4) and B (100% acetonitrile). The flow rate was maintained at 150 μL/min, with a 5 μL injection volume and a column temperature of 25 °C. Mass spectrometry was conducted in negative ion mode with a resolution of 140,000 at m/z 200, an automatic gain control (AGC) target of $1 \times 10^6$, and a scan range of m/z 75–1000. Data from isotope labeling experiments were processed using El-MAVEN, applying natural abundance correction.

### Mitochondrial electron transport chain complex activity assays

Complex II-IV activities were determined by Mitochondrial Complex Activity Assay Kit (complex II: BC3230, Solarbio; complex III: BC3240, Solarbio; complex IV: BC0945, Solarbio) followed the manufacturer's instructions using Visible spectrophotometer (GENESYS 30, Thermo Scientific).

### Oxygen consumption rate assay

The oxygen consumption rate (OCR) was measured using the Seahorse XFp Extracellular Flux Analyzer (Seahorse Bioscience). Experiments were performed according to the manufacturer's instructions. The Seahorse XF Mito stress test kit (Agilent Technologies) was used to test OCR by sequential injection of 50 μM oligomycin, 10 μM FCCP, and 2.5 μM rotenone/antimycin A.

### Measurement of cellular FAD levels

The cellular FAD levels were measured using the Flavin Adenine Dinucleotide (FAD) Assay Kit (Abcam, ab204710). Cells ($1 \times 10^6$) were harvested and resuspended in 400 μL ice-cold FAD Assay Buffer, then homogenized by pipetting. The homogenate was centrifuged at $10,000 \times g$ for 5 min at 4 °C to remove insoluble material. The supernatant was collected, and ice-cold 4 M Perchloric Acid (PCA) was added to a final concentration of 1 M, mixed briefly, and incubated on ice for 5 min. After centrifugation at $13,000 \times g$ for 2 min at 4 °C, the supernatant was transferred to a fresh tube, and the volume was measured. Ice-cold 2 M Potassium Hydroxide (KOH) was added to equal 34% of the supernatant volume, followed by vortexing and final centrifugation at $13,000 \times g$ for 15 min at 4 °C. The supernatant was then collected for FAD measurement.

### Cell proliferation and cell viability analyses

Cell proliferation assay (MTS) was performed using CellTiter 96_ kit (Promega, USA) as described in the manufacturer's instructions. The trypan blue exclusion assay (C0011, Beyotime) was performed according to the manufacturer's instructions.

### Collection of patient samples

To evaluate the clinical relevance of MT-CO2, primary human lung tumor specimens were collected from 23 male lung cancer patients (aged 48–72 years) who underwent surgical resection at Sichuan Provincial People's Hospital (Ethics Approval No. 2024-79-1). Samples were collected according to the institutional regulation of the Hospital Clinic Ethical Committee and were examined by pathologists. There were 11 lung tumor specimens harboring wild-type KRAS alleles ($KRAS^{WT}$) and 11 lung tumor specimens harboring KRAS mutant alleles ($KRAS^{MT}$). Tumor samples were subjected to qPCR analyses for MT-CO2, c-JUN, or GLS1 mRNA levels. No sex- or gender-based analyses were conducted in this study. All procedures were performed with signed informed consent from the patients, and the donation of lung tumor tissues was entirely voluntary and uncompensated.

### Statistics and reproducibility

GraphPad Prism 8.0 (GraphPad Software Inc., USA) was used for data recording, collection, processing, and calculation. Data from three independent experiments in vitro were presented as mean ± standard deviation (SD), and data from animal experiments were presented as mean ± standard error of the mean (SEM). Unpaired two-tailed Student's $t$ test was used for comparing two groups of data. One/two-way ANOVA with Tukey's test or Bonferroni's test was used to compare multiple groups of data. $P$ values ≤ 0.05 were considered significant. No statistical method was used to predetermine the sample size. No data were excluded from the analyses. Mice were randomly assigned to different experimental groups. Investigators measuring the mice were blinded to the treatment groups. Other experiments, such as the western blot assay, were conducted in a non-blinded manner.

### Reporting summary

Further information on research design is available in the Nature Portfolio Reporting Summary linked to this article.

## Data availability

The raw RNA sequencing and whole-exon sequencing data have been deposited in the NCBI Sequence Read Archive (SRA) under accession number PRJNA1002470. The mass spectrometry data for targeted metabolites and $^{13}C_5$-glutamine metabolic flux generated in this study are available in the Open Archive for Miscellaneous Data (OMIX) at the China National Center for Bioinformation / Beijing Institute of Genomics, Chinese Academy of Sciences, under accession codes OMIX008195 and OMIX008035. All data generated or analyzed during this study are included in this article and its Supplementary Information files. The uncropped gel or blot figures and original data underlying Figs. 1–7 and Supplementary Figs. S1–S12 are provided as a Source Data file. Source data are provided with this paper.

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

## Acknowledgements

We are grateful to Dr. Chong Chen (State Key Laboratory of Biotherapy, West China Hospital, Sichuan University, China) for the LSL-*KRAS*[G12D] knock-in mice. We thank members of the Z.-X.J.X. laboratory for stimulating discussions during the study. This work was supported in part by the National Natural Science Foundation of China (82073248) to Y.Y. and the National Key R&D Program of China (2022YFA1103700) to Z.-X.J.X.

## Author contributions

Y.Y., Z.-X.J.X., C.X., and Y.W. conceived and designed the research. Y.Y. and G.Q.W. performed most of the experiments with assistance from W.H.Z., S.H.Y., J.J.F., T.T.A., J.Q.Y, and J.Y. T.H., F.T.L., M.M.N. contributed to the data discussion. Y.Y. and Z.-X.J.X. wrote the manuscript. All the authors read and approved the manuscript.

## Competing interests

The authors declare no competing interests.
