## [Peer Review file · Nature Communications]

MT-CO2 promotes glutaminolysis and tumor cell survival through epigenetic regulation of glutaminase-1 upon glucose deprivation

Corresponding Author: Professor Zhi-Xiong Xiao

Version 0:

Reviewer comments:

Reviewer #1

(Remarks to the Author)

In this work, the authors describe a mechanism that support survival under glucose starvation.

A molecular cascade which involves Ras signaling activation, MT-CO2 transcription, histone-demethylase activation (KDM1A), Jun transcriptional activation and GLS1/glutaminolysis induction.

Of the above, the novelty of the study is the link of RAS activation to MT-CO2 transcription, and the downstream regulation of the epigenome. The general presented model is rather simplistic (glutaminolysis rescues from glucose starvation and stimulates tumorigenesis), but the authors did provide some mechanistic novelty to the process (though I question the specificity to MT-CO2).

1. The major pitfall of this study is the focus on MT-CO2. As a mtDNA-encoded protein, MT-CO2 is transcribed synchronously with other mtDNA-encoded genes (indeed in a TFAM dependent manner). Hence, the whole mechanism can be attributed to transcriptional regulation of mtDNA and not to MT-CO2 specifically. That said however, considering the essential components of NADH dehydrogenase that are co-transcribed with MT-CO2, it is hard to accept the suggested mechanism that complex IV is uniquely affected by glucose starvation, or uniquely induced in adapted cells. This required in-depth analyses of all mtDNA gene expression, and better analyses of mitochondrial complexes function (beyond the use of inhibitors). If indeed, complex I is also downregulated, the role of FAD may need to be revisited, or better analyzed.

2. Additionally to point 1 above, it will be required to demonstrate how cells survive glucose deprivation following effective knockdown or inhibition of other mtDNA encoded genes.

3. The introduction is very simplistic and inaccurate. It ignores many studies that rule out the relevance and importance of glutaminolysis in vivo, and the failure of GLS1 inhibitor as a clinical drug. Also, there is a contradiction in the statement that tumors are using more glucose and hence are glucose deprived. Obviously, the FDG-PET data from many clinical works argues against the availability of glucose to tumors. The definition of mitochondria as "semi-autonomous organelle" is unclear. Finally, it is expected that under glucose starvation oxphos will be required for energy, hence the essentiality of mtDNA is not completely unknown (suffice to discuss Rho-0 cells that depend not only on glucose, but also on pyruvate for extra-carbon source required for growth).

4. the abbreviation of glucose to Glu is confusing and wrong (Glu – glutamate), especially when it is mentioned alongside with the abbreviation of glutamine to Gln.

5. the role of GLS1 in oxphos under glucose starvation should be better explored. Does exogenous expression of GLS1 (or c-Jun) restore oxphos?

6. It is not obvious (or likely) that FAD enter cells when provided exogenously. The results in Figure 3H should be assessed alongside intracellular FAD levels.

7. While Ras signaling may stimulate MT-CO2, the biological and tumor growth consequences of MT-CO2 silencing are likely attributed to its housekeeping role. It is likely to silencing of any other essential component of oxphos will have a

similar effect on cell growth, especially under glucose starvation, and on tumor growth (whether Ras-induced or not).

Reviewer #2

(Remarks to the Author)

The authors present data that support a comprehensive pathway on how cancer cells can overcome the restriction of glucose in the tumor micro-environment and promote glutaminolysis to support their energy needs. They describe the role of MCO2, a component of mitochondrial ETC complex IV in the upregulation of the glutaminolysis pathway through activation of FAD driven effects on a nuclear demethylase that increases the expression of GLS1. They also show the direct role of RAS signaling pathway, (which is activated by either low glucose or by mutations), on TFAM induction, which goes to mitochondria to increase the expression of MCO2. Thus, their mechanism applies to all cancers showing the pro-survival mechanism of low glucose induced glutaminolysis, as well as the effects of that in pro-oncogenic signaling in tumors carrying RAS mutations.

The paper is very well written. It is clear and builds step by step in a very logical manner. The data are presented clearly and support all the conclusions. Its translational value is high. The correlations of their pathway with survival in cancer patients is very clear and identifies it as a therapeutic target. The described effects of berberine (a drug approved for other diseases) in animal xenotransplant models are solid. Because RAS inhibitor use is limited in patients with cancer, this makes the transition of berberine in clinical trials much easier.

I don't have any major concerns with the data.

The discussion is too long and distracting as most of it repeats the results and needs to be revised. It needs to discuss the new pathway in the context of what is known about how mTOR (which can actually sense glucose directly and also respond to glutamine directly) and AMPK regulate and respond to glutaminolysis (and RAS signaling).

This is important to show the independence of this pathway to what is known in the field, but I feel this paper has already a very large amount of nice data, so a nice discussion should suffice.

Reviewer #3

(Remarks to the Author)

In this manuscript authors evaluate the effects of glucose deprivation on the metabolic profile of lung adenocarcinoma, focusing on mitochondrial proteins encoded by the mtDNA. Upon adaptation to glucose limited conditions, H292 cells survive glucose starvation and have a higher tumorigenicity in vitro and in vivo. These adapted cells, called by the authors H292-V, reshape the mitochondrial encoded transcriptome, and in particular upregulate mtCO2, core component of complex IV. The authors then perform a large number of functional experiments to link elevated mtCO2 expression to enhanced glutaminolysis and to altered FAD levels, which impact the activity of histone de-methylase LSD1. These events are further linked together by the transcriptional regulator JUN, although the mechanism here seems a little twisted.

Overall, the study is interesting and tackles an hotly-debated issue in cancer metabolism. Experiments are articulated and well performed for the most part. Body of work is impressive. Some findings are obviously remarkable. Yet, I have some conceptual and technical concerns that substantially undermine conclusions.

A few conceptual concerns. First, authors clearly state that their goal is to understand mechanisms of resistance to glucose starvation. But their model is clearly designed to study mechanisms of adaptation to glucose limitation. These are two biologically different problems and authors should do a better job in parsing them out.

Along the same line, authors point to enhanced glutamine usage as a way to elude poor glucose availability. While this is a fairly accepted paradigm, more recent data in the field call for a re-evaluation of this theory. Glucose levels in the tumor interstitial fluid is generally sufficient for adequate ATP production and mitochondrial function, while cancer cells can scavenge several carbon sources from the tumor microenvironment for bioenergetics. In addition, if glucose levels were to be limiting, the same should be applied to glutamine and oxygen availability - two elements that might be critical factors in the study of mt-CO2 and are not reduced in the experiments presented in the manuscript.

While I do believe their findings have merit, authors should discuss better prior findings and have a broader perspective in the interpretation of results.

On another hand, authors claim the link between FAD and complex IV is well established. I don't think so, and references are missing. It is not clear to this reviewer how complex IV deficiency would impact FAD levels and/or regeneration, which is a pivotal point in the manuscript. In addition, the effect of de-methylation co-factors like α -KG is ignored.

More in general, it is not clear how much of the effects presented here are due to Complex IV activity and what is specific to mt-CO2. Complex IV is relatively understudied in cancer biology and insight would be welcome, but authors fail to provide a link between mt-CO2 elevation/disruption and Complex IV function.

There is another conceptual wrinkle I cannot entirely get. The FAD-LSD1-H3K9me2-c-JUN-GLS1 axis proposed in the manuscript implies that elevation of mtCO2 directly perturbs FAD levels and initiates a cascade of events that leads to increased GLS1 expression. However, authors seem to claim ("KRAS signaling" section) that sustained glutamine anaplerosis alters FAD levels. This creates a head-tail twist in the mechanism that I cannot completely grasp.

Specific points, major concerns:

- the effects of mtCO2 disruption on complex IV and mitochondrial biology were not documented. This is however very

important to 1) validate their KD system and 2) understand the mechanism of action. I recommend the authors perform full examination of mitochondrial respiration and function, together with analysis of Complex IV assembly.

- Data are inconclusive when it comes to glutaminolysis vs glutamine uptake. I recommend the authors perform isotope tracing analyses. Glutamine uptake should also be quantified.
- Mitochondrial mass cannot be assessed by Mitotracker. Authors should use citrate synthase abundance instead. In addition, the mitotracker they use is sensitive to mitochondrial TM potential, which makes their analysis even more misleading.
- It is not specified whether authors measure total FAD levels. If so, authors should also measure FAD+:FADH ratio. Similarly, it is not clear which form of FAD they provide to the cells in supplementation assays. As a side note, I was not aware of any FAD transporters on the plasma membrane; authors may want to briefly discuss how FAD enters the cell.
- Authors should take a deeper look at the effects of α -KG supplementation.
- Authors draw conclusions on the role of TFAM in mediating KRAS signaling, but no experiments actually interrogate TFAM. Please, either show experiments or re-state conclusions.
- Please, show experiments with acute MEK inhibition. Do those alter mt-CO2 at shorter time points? An argument against shRNA-type experiments is that cells are metabolically flexible over time and some rearrangements might be secondary to adaptation rather than direct signaling effectors.
- LSD1 is believed to act primarily on H3K4me2 and only marginally on H3K9me2 (if anything at all). While an effect by a related histone de-methylase is plausible, authors should show data on H3K4me2 if they focus on LSD1. Please, repeat ChIP experiments w/ H3K4me2.
- The experiments using berberine, and the conclusions authors draw are highly perplexing. Berberine is a recognized Complex I inhibitor, so parsing out the effects on Complex I vs Complex IV is hard, if not impossible. More in general, target specificity is a concern. Mitochondrial mass is measured only using Mitotracker, and that must be fixed. Rescue with FAD is not performed (rather, authors choose to rescue with α -KG for some reasons). Tumor data are OK, but authors should also show how treatment affects KrasWT tumors (to confirm specificity for KrasMUT tumors). To be honest, I thought this part did not add anything to the manuscript but rather undermined a little bit its overall value.

Minor issues:

- in Fig 1, one mitochondrial gene (mtATP8) is missing
- some terminology is funky, like "mass spectrum" or "transcriptional factors"
- Along the same point, the saying "H3K9me2 expression" is misleading as an histone mark is not technically expressed but a post-translational modification. Please, refer to H3K9me2 levels
- For RNA-Seq in Fig 3A, I would have appreciated a more unbiased approach to data analysis

Version 1:

Reviewer comments:

Reviewer #2

(Remarks to the Author)

The response of the author is satisfactory to both my comments, and, I believe, to the comment of the other reviewers. No further concerns.

Reviewer #3

(Remarks to the Author)

The authors addressed all my concerns, including conceptual points, through exhaustive experimentation and detailed explanation. I think the manuscript is significantly improved now and ready to be published in Nat Comms
Congratulations!

Rebuttal

Reviewer #1

Comments:

In this work, the authors describe a mechanism that support survival under glucose starvation. A molecular cascade which involves Ras signaling activation, MT-CO2 transcription, histone-demethylase activation (KDM1A), Jun transcriptional activation and GLS1/glutaminolysis induction. Of the above, the novelty of the study is the link of RAS activation to MT-CO2 transcription, and the downstream regulation of the epigenome. The general presented model is rather simplistic (glutaminolysis rescues from glucose starvation and stimulates tumorigenesis), but the authors did provide some mechanistic novelty to the process (though I question the specificity to MT-CO2).

Point 1:

The major pitfall of this study is the focus on MT-CO2. As a mtDNA-encoded protein, MT-CO2 is transcribed synchronously with other mtDNA-encoded genes (indeed in a TFAM dependent manner). Hence, the whole mechanism can be attributed to transcriptional regulation of mtDNA and not to MT-CO2 specifically. That said however, considering the essential components of NADH dehydrogenase that are co-transcribed with MT-CO2, it is hard to accept the suggested mechanism that complex IV is uniquely affected by glucose starvation, or uniquely induced in adapted cells. This required in-depth analyses of all mtDNA gene expression, and better analyses of mitochondrial complexes function (beyond the use of inhibitors). If indeed, complex I is also downregulated, the role of FAD may need to be revisited, or better analyzed.

Response 1: The specific effect on MT-CO2 is indeed a vital issue of this study. We sincerely appreciate the reviewer's insightful comments and constructive suggestions. MT-CO2 and other 11 respiratory chain complex subunits are encoded by the H-strand on the mitochondria genome. Thus, the 12 subunits should theoretically have the same copy numbers of the respective mRNAs. However, our data indicates that glucose deprivation robustly elevated the mRNA levels of MT-CO2 but not other mt-mRNAs, including MT-ND1, MT-ND5, and MT-ATP6 (Revised Fig.1g), suggesting that MT-CO2 mRNA stability is most likely modulated.

Accumulating evidence indicates that N6-methyladenosine (m6A) modification plays a critical role in modulating mRNA stability. Currently, there is no report about m6A modification on mitochondrial mRNAs. We thus investigated the possibility that MT-CO2 mRNA may be modified by m6A. As the SRAMP database (<http://www.cuilab.cn/sramp>) predicted, several MT-mRNAs, including MT-CO2, possess putative m6A modifications (Revised supplemental table 2). We then performed a set of new experiments.

Immunofluorescence analyses showed that m6A was co-localized with MT-

CO2 (Rebuttal Fig.1a-b, attached below). In addition, RNA immunoprecipitation (RIP) analyses using m6A-specific antibodies indicated that MT-CO2 mRNA was m6A modified, apparently more than other mt-mRNAs (Rebuttal Fig.1c). Notably, RNAseq and Western blot analyses show that glucose starvation-resistant H292-V cells had lower IGF2BP3 levels, a documented m6A reader (*Zaccara et al., Nat Rev Mol Cell Biol., 2019, PMID: 31520073*), than parental H292 cells at both mRNA and protein levels (Rebuttal Fig.1d-e). Furthermore, immunofluorescence analyses showed that endogenous as well as exogenous IGF2BP3 could be localized in mitochondria (Rebuttal Fig.1f-i). These results suggest that IGF2BP3 could regulate MT-CO2 expression. Indeed, silencing IGF2BP3 specifically led to upregulating the MT-CO2 mRNA and protein levels (Rebuttal Fig.1j-k). Conversely, ectopic expression of IGF2BP3 remarkably reduced MT-CO2 mRNA and protein levels (Rebuttal Fig.1l-m).

We further dissected the molecular mechanism by which IGF2BP3 modulates the expression of MT-CO2 mRNA. As shown in Rebuttal Fig. 2a, the knockdown of IGF2BP3 significantly upregulated MT-CO2 mRNA stability. In contrast, ectopic expression of IGF2BP3 notably decreased MT-CO2 mRNA stability (Rebuttal Fig. 2b). Moreover, RIP analyses showed that IGF2BP3 could bind to MT-CO2 mRNA but not to other mt-mRNAs, such as MT-CYB, MT-ND1, MT-ND3, MT-ND4L, or MT-CO1 (Rebuttal Fig.2c). Importantly, restoration of IGF2BP3 in H292-V cells markedly reduced MT-CO2 expression to levels similar to those in parental H292 cells (Rebuttal Fig.2d).

Together, these new results indicate that IGF2BP3 preferentially binds to and inhibits MT-CO2 mRNA stability. Given that glucose starvation facilitates the transcription of MT-CO2 and other MT-mRNAs by activating Ras signaling, the glucose starvation-mediated downregulation of IGF2BP3 most likely accounts for the specific upregulation of MT-CO2 mRNA stability. These new results have been integrated into revised Fig.5 and S10.

Rebuttal Fig.1. (a-b) m6A was co-localized with MT-CO2. (c) MT-CO2 mRNA was m6A modified, apparently more than other mt-mRNAs. (d-e) H292-V cells had lower IGF2BP3 levels than parental H292 cells. (f-i) IGF2BP3 could be localized in mitochondria. (j-k) Silencing IGF2BP3 led to upregulating the MT-CO2 mRNA and protein levels. (l-m) Ectopic expression of IGF2BP3 reduced MT-CO2 mRNA and protein levels.

Rebuttal Fig.2. (a) knockdown of IGF2BP3 upregulated MT-CO2 mRNA stability. (b) Ectopic expression of IGF2BP3 decreased MT-CO2 mRNA stability. (c) IGF2BP3 could bind to MT-CO2 mRNA but not to other mt-mRNAs, such as MT-CYB, MT-ND1, MT-ND3, MT-ND4L, or MT-CO1. (d) Restoration of IGF2BP3 in H292-V cells reduced MT-CO2 expression to levels.

Regarding the role of the respiratory complexes in the elevation of FAD.

To address this issue, we performed a new set of experiments. First, we examined the cellular FAD levels upon specifically knocking down a key mitochondria-encoded protein in the respiratory complexes, including MT-ND1 (a subunit of complex I), MT-CYB (a subunit of complex III), MT-CO2 (a subunit of complex IV), or MT-ATP6 (a subunit of complex V). Notably, previous reports show that inhibition of complex II-mediated electron transport suppresses the regeneration of FAD from FADH₂ (Erich Gnaiger, *JBC*, 2024, PMID: 38118236; Hou et al., *Appl Microbiol Biotechnol.*, 2023, PMID: 37354265). Our results showed that the knockdown of MT-CYB, MT-CO2, or MT-ATP6 but not MT-ND1 significantly reduced cellular FAD levels (Rebuttal Fig.3a), indicating that inhibition of complexes II-V disrupts electron transport, leading to reduced regeneration of FAD from FADH₂.

It is reported that inhibition of mitochondrial complex IV leads to a secondary

loss of complexes II and III activity (*Hargreaves et al., Mitochondrion, 2007, PMID: 17395552; LaMoia et al., PNAS, 2022, PMID: 35238637*), thereby further suppressing FAD regeneration from FADH₂. Indeed, in our new results, silencing of MT-CO2 significantly suppressed both the cellular oxygen consumption rate (OCR) and the activity of complexes II, III, and IV (Rebuttal Fig. 3b and 3i).

We previously showed that pharmacological inhibition of respiratory chain complex II, III, IV, or V but not complex I significantly suppressed c-JUN expression (Revised Fig.S4a-f). During revision, we further investigated the effects of silencing each respiratory complex on the expression of c-JUN and GLS1. Silencing of MT-ND1 led to reduced OCR as expected, but had little impact on c-JUN and GLS1 expression (Rebuttal Fig.3c and 3g). By sharp contrast, the knockdown of either MT-CO2, MT-CYB, or MT-ATP6 led to reduced OCR and significantly suppressed both c-JUN and GLS1 expression (Rebuttal Fig.3d-f and 3h-j). Notably, the knockdown of MT-CO2, MT-CYB, or MT-ATP6 but not MT-ND1 led to the remarkable elevation of H3K9me2, a marker of LSD1 inactivation (Rebuttal Fig.3c-f).

Together, these results indicate that disruption of complexes II-IV, exemplified by the knockdown of MT-CO2 exclusively studied in this study, downregulates the FAD levels, consequently suppressing the LSD1 activity, GLS1 expression, and glutaminolysis. Consistently, glucose deprivation-mediated upregulation of MT-CO2 expression results in the elevation of FAD and activation of LSD1, consequently facilitating c-JUN-mediated GLS1 expression and glutaminolysis. These new results were integrated into the revised Fig.3f-g and S4g-m.

Rebuttal Fig.3. Silencing of MT-CO2, MT-CYB, or MT-ATP6 but not MT-ND1 led to reduced FAD levels and significantly suppressed both c-JUN and GLS1 expression.

Point 2:

Additionally to point 1 above, it will be required to demonstrate how cells survive glucose deprivation following effective knockdown or inhibition of other mtDNA encoded genes.

Response 2: We sincerely appreciate your suggestions. In response, we show that the knockdown of MT-CYB or MT-ATP6, similar to the silencing of MT-CO2, significantly suppressed H1299 cell viability under Glc-/Gln+ condition (DMEM containing 2mM glutamine in the absence of glucose) (Rebuttal Fig.4a-b). Consistently, pharmacologic inhibition of respiratory chain complexes II, III, IV, and V by the respective inhibitor (carboxin for complex II, antimycin for complex III, NaN₃ for complex IV, and oligomycin for complex V) also dramatically suppressed H1299 cell viability under Glc-/Gln+ condition (Rebuttal Fig.4c-d). These results indicate that the inhibition of respiratory complexes not only blocks oxidative phosphorylation but also blocks glutaminolysis, consequently resulting in tumor cell death upon glucose deprivation. Indeed, a supplement with α -ketoglutarate (α -KG), a metabolic intermediate of glutaminolysis, could largely rescue tumor cell viability inhibited by silencing MT-CO2 upon Glc-/Gln+

condition (Revised Fig.2g-i).

Rebuttal Fig. 4. Inhibition of the respiratory chain complexes II-V suppresses tumor cell viability upon glucose deprivation.

Point 3:

The introduction is very simplistic and inaccurate. It ignores many studies that rule out the relevance and importance of glutaminolysis in vivo, and the failure of GLS1 inhibitor as a clinical drug. Also, there is a contradiction in the statement that tumors are using more glucose and hence are glucose deprived. Obviously, the FDG-PET data from many clinical works argues against the availability of glucose to tumors. The definition of mitochondria as “semi-autonomous organelle is unclear. Finally, it is expected that under glucose starvation oxphos will be required for energy, hence the essentiality of mtDNA is not completely unknown (suffice to discuss Rho-0 cells that depend not only on glucose, but also on pyruvate for extra-carbon source required for growth).

Response 3: The relevance and importance of glutaminolysis *in vivo* tumor growth is indeed an important issue.

Abundant evidence demonstrates that glutaminolysis is crucial in tumor growth *in vivo* (Altman et al., *Nature Review Cancer*, 2016, PMID: 28704361). For instance, mice with a 50% allelic reduction of GLS1 significantly delay early onset tumor progression in the MYC-induced liver cancer model (Xiang et al., *J Clin Invest.*, 2015, PMID: 25915584). GLS1 heterozygous knockout mice exhibit significant elevation of radiosensitivity in a P7NP (*Pax7^{Cre-ER-T2/+}*,

Nras^{LSL-G12D/+; p53^{fl/fl}} sarcoma mouse model (Patel et al., *Communications Biology*, 2024, PMID: 38769385). Several xenograft mouse models demonstrate significant retardation of tumor growth derived from GLS1 knocking-down tumor cells, including hepatocellular carcinoma (Li et al., *EBioMedicine*, 2024, PMID: 30555042), glioblastoma (Cheng et al., *PNAS*, 2011, PMID: 21555572), and colon cancer (Xiang et al., *CDDis*, 2019, PMID: 30674873). Similarly, pharmacological inhibition of GLS1 with CB-839 or BPTES substantially suppressed tumor growth *in vivo* (Le et al., *Cell Metabolism*, 2012, PMID: 22225880; Wu et al., *Nature Cancer*, 2021, PMID: 34085048). In addition, the inhibition of SLC1A5, a key glutamine transporter, can reduce xenograft NSCLC or HNSCC tumor growth (Hassanein et al., *International Journal of Cancer*, 2015, PMID: 25821004; Zhang et al., *British Journal of Cancer*, 2019, PMID: 31819178).

Despite previous reports that CB-839 failed to suppress KRAS-driven pancreatic ductal adenocarcinoma tumor growth in a xenograft mouse model (Biancur et al., *Nature Communications*, 2017, PMID: 28671190) and was unable to demonstrate benefit over standard-of-care immunotherapy in patients with stage IV NSCLC (NCT04265534) (Vasan et al., *Journal of Clinical Investigation*, 2024, PMID: 38299592), a recent phase 1b/2 trial indicated that a combination of CB-839 and azacytidine is a safe and effective approach for treating myelodysplastic syndrome (NCT03047993) (DiNardo et al., *Nature Cancer*, 2024, PMID: 39300320). Furthermore, a phase II trial showed that combining CB-839 with everolimus significantly improved progression-free survival in patients with renal cell carcinoma (NCT03163667) (Lee et al., *Clinical Cancer Research*, 2022, PMID: 35576438). These recent developments highlight the therapeutical potential of targeting glutaminolysis when combined with an appropriate drug(s). Indeed, several ongoing clinical trials are investigating CB-839 in combination with capecitabine (NCT02861300), sapanisertib (NCT04250545), osimertinib (NCT03831932), and temozolomide (NCT03528642) for various cancers. In the revised manuscript, we have integrated the above discussion (Lines 577-587).

“Also, there is a contradiction in the statement that tumors are using more glucose and hence are glucose deprived. Obviously, the FDG-PET data from many clinical works argues against the availability of glucose to tumors”.

FDG-PET/CT scans were developed for clinical tumor detection using FDG (¹⁸F-2-deoxyglucose) as a marker of tumor cells, which can take in FDG but cannot metabolize it, resulting in the accumulation of FDG in tumor cells. Therefore, FDG levels can reflect the ability of tumor cells to ingest glucose, yet they do not reflect the glucose levels in the tumor microenvironment. In fact, it has been shown that glucose concentrations in solid tumor tissues are 3- to 10-fold lower than in normal tissues (Hirayama et al., *Cancer Res.*, 2009, PMID: 19458066; Birsoy et al., *Nature*, 2014, PMID: 24670634), indicating that

glucose deprivation is one of the characteristics of the tumor microenvironment.

“The definition of mitochondria as semi-autonomous organelle is unclear”.

In response, we removed this phrase in the revised manuscript lines 79-80.

“Finally, it is expected that under glucose starvation oxphos will be required for energy, hence the essentiality of mtDNA is not completely unknown (suffice to discuss Rho-0 cells that depend not only on glucose, but also on pyruvate for extra-carbon source required for growth)”.

Our results demonstrate that inhibition of respiratory complexes, including the knockdown of mtDNA-encoded genes and pharmacological inhibitors, results in tumor cell death upon glucose starvation (Rebuttal Fig.4), indicating that oxidative phosphorylation is essential for tumor cell viability upon glucose starvation.

Rho-0 cells bearing mtDNA loss due to ethidium bromide (EB) induction depend on glucose for survival (*Haga et al., Cancer Science, 2010, PMID: 20210797*). In our new set of experiments, EB-induced loss of mtDNA resulted in A549 cell death under the Glc-/Gln+ condition (Rebuttal Fig.5a-b), indicating that glucose is indispensable for cell viability upon loss of mtDNA. Since mitochondria are essential for converting pyruvate into ATP via oxidative phosphorylation, supplementing glucose, but not pyruvate, could rescue the viability of mtDNA-deficient A549 cells under Glc-/Gln+ condition (Rebuttal Fig. 5c). These new results were included in the revised Fig.S1d-e.

Rebuttal Fig. 5. (a-b) Mitochondrial DNA loss induced by ethidium bromide results in A549 cell death upon Glc-/Gln+ condition. (c) Supplementing glucose, but not pyruvate, could rescue the viability of mtDNA-deficient A549 cells under the Glc-/Gln+ condition.

Point 4:

the abbreviation of glucose to Glu is confusing and wrong (Glu – glutamate),

especially when it is mentioned alongside with the abbreviation of glutamine to Gln.

Response 4: We apologize for the confusion. In the revised manuscript, we have used the abbreviation "Glc" for glucose.

Point 5:

the role of GLS1 in oxphos under glucose starvation should be better explored. Does exogenous expression of GLS1 (or c-Jun) restore oxphos?

Response 5: Thank you for your suggestion. Due to the technical limitation, oxygen consumption rate (OCR), which reflects oxidative phosphorylation (OXPHOS), cannot be measured without glucose. We therefore examined the effect of ectopic GLS1 expression in OXPHOS. As shown in Rebuttal Fig.6a-b, ectopic expression of GLS1 significantly increased the OCR levels in H292 cells, indicating that GLS1 plays a critical role in promoting OXPHOS. These new results were integrated into the revised Fig.S2e-f.

Rebuttal Fig.6. Ectopic expression of GLS1 increases OCR levels in H292 cells.

Point 6:

It is not obvious (or likely) that FAD enter cells when provided exogenously. The results in Figure 3H should be assessed alongside intracellular FAD levels.

Response 6: Thank you for your comment and suggestion. Indeed, there are currently no reports indicating the existence of an FAD transporter in the plasma membrane. However, our results demonstrate that supplement with FAD (dissolved in DMSO) in the media significantly increased intracellular FAD levels in a dose-dependent manner (Rebuttal Fig.7a), in keeping with a previous report (Hirano *et al.*, *FEBS Open Bio.*, 2017, PMID: 29226080). Notably, FAD treatment could markedly restore both cellular FAD levels and the c-JUN/GLS1 protein expression inhibited by silencing of MT-CO2 (Rebuttal Fig.7b-c).

In the cell, FAD is synthesized from riboflavin, which is transported into the cell by its specific transporter. Our results show that, similar to FAD, treatment with riboflavin also significantly increased intracellular FAD levels in a dose-dependent manner (Rebuttal Fig.7d). Furthermore, riboflavin treatment markedly restored cellular FAD levels and the c-JUN/GLS1 protein expression, inhibited by the silencing of MT-CO2 (Rebuttal Fig.7e-f).

Together, these results further indicate that the knockdown of MT-CO2 leads to reduced FAD levels, resulting in the downregulation of c-JUN and GLS1 expression. These new results were integrated into the revised Fig.3l-o and S4n-o.

Rebuttal Fig.7. FAD (a-c) or riboflavin (d-f) treatment could markedly restore both cellular FAD levels and the c-JUN/GLS1 protein expression inhibited by silencing of MT-CO2.

Point 7:

While Ras signaling may stimulate MT-CO2, the biological and tumor growth consequences of MT-CO2 silencing are likely attributed to its housekeeping role. It is likely to silencing of any other essential component of oxphos will have a similar effect on cell growth, especially under glucose starvation, and on tumor growth (whether Ras-induced or not).

Response 7: These points are very important. Indeed, mitochondria DNA is essential for KRAS-driven tumor growth *in vivo* (Weinberg *et al.*, PNAS, 2010, PMID: 20421486). Consistently, our results show that, similar to the knockdown of MT-CO2, silencing of MT-ND1 (a subunit of complex I), MT-CYB (a subunit of complex III), or MT-ATP6 (a subunit of complex V) also dramatically suppressed H1299 (Ras mutant) cell proliferation (Rebuttal Fig.8), supporting

the notion that a housekeeping role of MT-CO2 and other essential components in the respiratory chain for RAS-driven tumor growth.

However, in response to glucose deprivation, MT-CO2 expression is robustly upregulated, achieved by Ras signaling-mediated transcriptional activation and IGF2BP3-mediated modulation of MT-CO2 mRNA stability (Rebuttal Fig.1 and Revised Fig.4). These results provide strong evidence that elevation of MT-CO2 is essential for tumor cell survival in response to glucose shortage.

Rebuttal Fig. 8. Silencing of MT-ND1, MT-CYB, MT-CO2, or MT-ATP6 suppresses H1299 cell proliferation.

Reviewer #2

Comments:

The authors present data that support a comprehensive pathway on how cancer cells can overcome the restriction of glucose in the tumor micro-environment and promote glutaminolysis to support their energy needs. They describe the role of MCO2, a component of mitochondrial ETC complex IV in the upregulation of the glutaminolysis pathway through activation of FAD driven effects on a nuclear demethylase that increases the expression of GLS1. They also show the direct role of RAS signaling pathway, (which is activated by either low glucose or by mutations), on TFAM induction, which goes to mitochondria to increase the expression of MCO2. Thus, their mechanism applies to all cancers showing the pro-survival mechanism of low glucose induced glutaminolysis, as well as the effects of that in pro-oncogenic signaling in tumors carrying RAS mutations.

The paper is very well written. It is clear and builds step by step in a very logical manner. The data are presented clearly and support all the conclusions. Its translational value is high. The correlations of their pathway with survival in cancer patients is very clear and identifies it as a therapeutic target. The described effects of berberine (a drug approved for other diseases) in animal xenotransplant models are solid. Because RAS inhibitor use is limited in patients with cancer, this makes the transition of berberine in clinical trials much easier.

I don't have any major concerns with the data.

The discussion is too long and distracting as most of it repeats the results and needs to be revised. It needs to discuss the new pathway in the context of what is known about how mTOR (which can actually sense glucose directly and also respond glutamine directly) and AMPK regulate and respond to glutaminolysis (and RAS signaling).

This is important to show the independence of this pathway to what is known in the field, but I feel this paper has already a very large amount of nice data, so a nice discussion should suffice.

Response: We sincerely appreciate the reviewer's positive comments and insightful suggestions. As suggested, we revised the discussion to be more focused. We also included a brief discussion on the role of the AMPK/mTOR pathway in tumor cell survival under glucose deprivation and the potential connection between the AMPK/mTOR pathway and the new pathway involved in the Ras-MT-CO2-glutaminolysis axis (Revised manuscript lines 533-555).

Reviewer #3

Comments:

In this manuscript authors evaluate the effects of glucose deprivation on the metabolic profile of lung adenocarcinoma, focusing on mitochondrial proteins encoded by the mtDNA. Upon adaptation to glucose limited conditions, H292 cells survive glucose starvation and have a higher tumorigenicity in vitro and in vivo. These adapted cells, called by the authors H292-V, reshape the mitochondrial encoded transcriptome, and in particular upregulate mtCO2, core component of complex IV. The authors then perform a large number of functional experiments to link elevated mtCO2 expression to enhanced glutaminolysis and to altered FAD levels, which impact the activity of histone de-methylase LSD1. These events are further linked together by the transcriptional regulator JUN, although the mechanism here seems a little twisted.

Overall, the study is interesting and tackles an hotly-debated issue in cancer metabolism. Experiments are articulated and well performed for the most part. Body of work is impressive. Some findings are obviously remarkable. Yet, I have some conceptual and technical concerns that substantially undermine conclusions.

A few conceptual concerns. First, authors clearly state that their goal is to understand mechanisms of resistance to glucose starvation. But their model is clearly designed to study mechanisms of adaptation to glucose limitation. These are two biologically different problems and authors should do a better job in parsing them out.

Response: We sincerely appreciate the reviewer's comments and suggestions. Indeed, our study aimed to elucidate how tumor cells sense and adapt to glucose deprivation in the tumor microenvironment. We demonstrate that glucose deprivation-induced upregulation of MT-CO2 plays a critical role in promoting glutaminolysis in overcoming glucose shortage. We have made this point more explicit throughout the revised manuscript.

Along the same line, authors point to enhanced glutamine usage as a way to elude poor glucose availability. While this is a fairly accepted paradigm, more recent data in the field call for a re-evaluation of this theory. Glucose levels in the tumor interstitial fluid is generally sufficient for adequate ATP production and mitochondrial function, while cancer cells can scavenge several carbon sources from the tumor microenvironment for bioenergetics. In addition, if glucose levels were to be limiting, the same should be applied to glutamine and oxygen availability - two elements that might be critical factors in the study of mt-CO2 and are not reduced in the experiments presented in the manuscript. While I do believe their findings have merit, authors should discuss better prior findings

and have a broader perspective in the interpretation of results.

Response: We sincerely appreciate your comments and suggestions. Indeed, in addition to glucose and glutamine, cancer cells can uptake several carbon sources from the tumor microenvironment for bioenergetics, including acetate and lactate (Hui *et al.*, *Nature*, 2017, PMID: 29045397; Schug *et al.*, *Nat Rev Cancer.*, 2016, PMID: 27562461). Moreover, fatty acids and glycogen can also serve as alternative energy sources for cancer cells. To directly address this issue, our new results presented in the revised manuscript show that inhibition of glutaminolysis by CB-839 (a GLS1 inhibitor), but not inhibition of fatty acid oxidation by etomoxir (a CPT1a inhibitor) or inhibition of glycogen catabolism by CP-91149 (a glycogen phosphorylase inhibitor), dramatically suppressed the survival of A549 and H1299 cells under Glc-/Gln+ condition (DMEM containing 2mM glutamine in the absence of glucose) (Rebuttal Fig.9a-d). Notably, ectopic expression of GLS1, alone, led to H292 cell survival under Glc-/Gln+ condition and tumor growth *in vivo* (Rebuttal Fig.9e-i). These results indicate that glutamine/glutaminolysis is essential for cancer cell survival under glucose deprivation.

It has been reported that the glucose concentrations in the tumor tissues are 3- to 10-fold lower than in normal tissues (Hirayama *et al.*, *Cancer Res.*, 2009, PMID: 19458066; Birsoy *et al.*, *Nature*, 2014, PMID: 24670634). However, it has been reported that glutamine, unlike glucose, is not deprived in the tumor microenvironment (Hirayama *et al.*, *Cancer Res.*, 2009, PMID: 19458066). A recent report shows that glutamine is a primary carbon source to be catabolized in mitochondria, ahead of fatty acids, thereby compensating for glucose scarcity under starvation (Li *et al.*, *Cell Res.*, 2024, PMID: 38898113).

Together, our results strongly support the notion that glutamine is the preferred energy source for tumor cells upon glucose deprivation. We have integrated these new results into the revised Fig. S2a-i.

Rebuttal Fig.9. (a-d) Inhibition of GLS1 results in cell death under the Glc-/Gln+ condition. (e-i) GLS1 promotes tumor cell survival under Glc-/Gln+ condition and xenograft tumor growth *in vivo*.

“On another hand, authors claim the link between FAD and complex IV is well established. I don’t think so, and references are missing. It is not clear to this reviewer how complex IV deficiency would impact FAD levels and/or regeneration, which is a pivotal point in the manuscript. In addition, the effect of de-methylation co-factors like a-KG is ignored”.

Response: Thank you for your comments. FADH₂, derived from the TCA cycle or fatty acid β-oxidation in the mitochondria matrix, deposits electrons at complex II, converting into FAD and releasing 2 H⁺. Inhibition of complex II-mediated electron transport suppresses the regeneration of FAD from FADH₂ (Erich Gnaiger, *JBC*, 2024, PMID: 38118236; Hou et al., *Appl Microbiol Biotechnol.*, 2023, PMID: 37354265). Notably, inhibition of mitochondrial complex IV leads to a secondary loss of complexes II–III activity (Hargreaves et al., *Mitochondrion*, 2007, PMID: 17395552; LaMoia et al., *PNAS*, 2022, PMID: 35238637), thereby further suppressing FAD regeneration from FADH₂. In keeping with this observation, our results show that the knockdown of MT-CO2 (a subunit of complex IV), MT-CYB (a subunit of complex III), or MT-ATP6 (a subunit of complex V) significantly reduced cellular FAD levels (Rebuttal Fig.

10a). In addition, silencing of MT-CO2 markedly suppressed the activity of complexes II, III, and IV (Rebuttal Fig.10b). Furthermore, compared to parental H292 cells, glucose starvation-resistant H292 cells (H292-V) exhibited higher MT-CO2 levels and increased activity of complexes II, III, and IV (Rebuttal Fig. 10c and Revised Fig.1g-h). Moreover, ectopic expression of KRas^{G12V} significantly upregulated the expression of MT-CO2 concomitant with increased activity of respiratory chain complexes II-IV and FAD levels (Revised Fig.4c and Rebuttal Fig.10d-e).

Together, these results demonstrate that glucose starvation-mediated elevation of MT-CO2 upregulates complexes II-IV activity, resulting in FAD regeneration, hence providing a link between FAD and complex IV. We have included the relevant references in the revised manuscript (Lines 254-255) and integrated new results into the revised Fig.3g, 3i, S4m, S7a, and S7e.

Rebuttal Fig.10. (a-b) Silencing of MT-CO2 markedly suppressed the activity of complexes II-IV and reduced cellular FAD levels. (c) Compared to parental H292 cells, H292-V exhibited increased complexes II-IV activity. (d-e) Ectopic expression of KRas^{G12V} significantly increased the activity of respiratory chain complexes II-IV and FAD levels.

In response to the suggestion of using α -KG to explore its role in the MT-CO2 knockdown-mediated epigenetic regulation, we performed new experiments by supplementation with α -KG. Our results show that α -KG could rescue tumor cell viability inhibited by the silencing of MT-CO2 under Glc-/Gln condition (Rebuttal Fig.11a), whereas it had little effect on the levels of

H3K9me2 and H3K4me2 upregulated by silencing of MT-CO2 (Rebuttal Fig.11b), indicating that α -KG serves as an intermediate of glutaminolysis to rescue tumor cell survival upon glucose deprivation and that α -KG does not impact MT-CO2-mediated epigenetic alternation. This new result was integrated into revised Fig.2h and S5b.

Rebuttal Fig. 11. α -KG has little effect on the levels of H3K9me2 and H3K4me2 upregulated by the knockdown of MT-CO2.

“More in general, it is not clear how much of the effects presented here are due to Complex IV activity and what is specific to mt-CO2. Complex IV is relatively understudied in cancer biology and insight would be welcome, but authors fail to provide a link between mt-CO2 elevation/disruption and Complex IV function”.

Response: MT-CO2 is a core subunit of complex IV, essential for the assembly and activity of complex IV (Courtois et al., BBA, 2024, PMID: 37640115). As shown in the Rebuttal Fig.12a-b, silencing of MT-CO2 suppressed the activity of complex IV, accompanied by reduced oxygen consumption rate (OCR). Consistently, glucose starvation-resistant H292 cells (H292-V) exhibited higher MT-CO2 levels, concomitant with increased activity of complex IV and increased OCR levels (Rebuttal Fig.12c-d and Revised Fig.1g-h). In addition, ectopic expression of KRas^{G12V} significantly upregulated the expression of MT-CO2 concomitant with increased activity of respiratory chain complex IV and OCR levels (Rebuttal Fig.12e-f and Revised Fig.4c). These results indicate that there is a strong link between MT-CO2 elevation/disruption and complex IV function.

Notably, there is an intrinsic connection among the complexes. It is known that the inhibition of mitochondrial complex IV leads to a secondary loss of complexes II–III activity (Hargreaves et al., Mitochondrion, 2007, PMID: 17395552; LaMoia et al., PNAS, 2022, PMID: 35238637). In keeping with this line, our results show that the knockdown of MT-CO2 significantly suppressed not only the activity of complex IV but also complexes II and III (Rebuttal Fig.12a). Together, our results indicate that glucose deprivation leads to robust elevation of MT-CO2, resulting in the upregulation of activity of complexes II-IV. This cascade leads to the elevation of FAD levels and the epigenetic

upregulation of GLS1, ultimately promoting glutaminolysis. These new results were integrated into the revised Figure 3f-i and S7a-b.

Rebuttal Fig.12. (a-b) Silencing of MT-CO2 suppresses the activity of complexes II-IV and OCR levels. (c-d) Compared to parental H292 cells, H292-V cells exhibited increased activity of complexes II-IV and OCR levels. (e-f) Ectopic expression of KRas^{G12V} increased activity of respiratory chain complexes II-IV and OCR levels.

“There is another conceptual wrinkle I cannot entirely get. The FAD-LSD1-H3K9me2-c-JUN-GLS1 axis proposed in the manuscript implies that elevation of mtCO2 directly perturbs FAD levels and initiates a cascade of events that leads to increased GLS1 expression. However, authors seem to claim (“KRAS signaling” section) that sustained glutamine anaplerosis alters FAD levels. This creates a head-tail twist in the mechanism that I cannot completely grasp”.

Response: Thank you for the comments. In this study, we show that activation of Ras signaling, either by glucose starvation or ectopic expression of KRas^{G12V}, leads to upregulated MT-CO2 expression, which enhances complex IV activity and elevates FAD levels in activating LSD1. Activated LSD1 subsequently upregulates JUN transcription by reducing H3K9me2 levels, leading to increased GLS1 expression and enhanced glutaminolysis.

Regarding the issue of the sustained glutamine anaplerosis altering FAD levels, our results show that silencing of MT-CO2 significantly inhibited FAD levels in the presence of glucose and glutamine. Furthermore, we show that activation of Ras promoted glutaminolysis, accompanied by elevated FAD levels, which was inhibited by silencing of MT-CO2 (Revised Fig.S7), suggesting that MT-CO2, not glutamine, is primarily responsible for altering

FAD levels.

Specific points, major concerns:

Point 1:

-the effects of mtCO2 disruption on complex IV and mitochondrial biology were not documented. This is however very important to 1) validate their KD system and 2) understand the mechanism of action. I recommend the authors perform full examination of mitochondrial respiration and function, together with analysis of Complex IV assembly.

Response 1: Thank you for your comments and suggestions. In response, we investigated the impact of MT-CO2 silencing on complex IV activity and cellular oxygen consumption rates (OCR). As shown in Rebuttal Fig.12a, silencing MT-CO2 significantly reduced complex IV activity, accompanied by decreased activities of complexes II and III. Furthermore, MT-CO2 knockdown led to a reduction in cellular OCR levels (Rebuttal Fig.12b). Notably, previous studies have reported that MT-CO2 is critical for complex IV assembly (*Mukherjee et al., Mitochondrion, 2020, PMID: 32304865*). These findings indicate that MT-CO2 is crucial for maintaining complex IV activity and overall mitochondrial respiration.

Point 2:

- Data are inconclusive when it comes to glutaminolysis vs glutamine uptake. I recommend the authors perform isotope tracing analyses. Glutamine uptake should also be quantified.

Response 2: Thank you for the comments and suggestions. Our previous mass spectrometry analyses show that the silencing of MT-CO2 in H1299 cells dramatically increased cellular glutamine levels, possibly due to increased glutamine uptake or inhibited glutaminolysis. To address this issue, we examined the effects of MT-CO2 in the expression of glutamine transporter SLC1A5. As shown in Rebuttal Fig.13a, the knockdown of MT-CO2 had little effect on SLC1A5 expression, suggesting that silencing of MT-CO2 likely leads to a blockage of glutaminolysis. Using mass spectrometry, we traced the metabolic flux of $^{13}\text{C}_5$ -glutamine. As shown in rebuttal Fig.13b, there was a reduced ^{13}C fraction from $^{13}\text{C}_5$ -glutamine in downstream metabolites, including glutamate, α -ketoglutarate, succinate, fumarate, and citrate. Additionally, $^{13}\text{C}_5$ -glutamine was accumulated upon knockdown of MT-CO2, indicating that silencing MT-CO2 inhibits glutaminolysis, preventing tumor cells from utilizing glutamine. These new results were integrated into the revised Fig. 2d and S2k.

Rebuttal Fig.13. (a) Silencing of MT-CO2 has little effect on SLC1A5. (b) Silencing MT-CO2 inhibits glutaminolysis.

Point 3:

- Mitochondrial mass cannot be assessed by Mitotracker. Authors should use citrate synthase abundance instead. In addition, the mitotracker they use is sensitive to mitochondrial TM potential, which makes their analysis even more misleading.

Response 3: Thank you for the insightful comments and suggestions. As suggested, we examined the citrate synthase expression in our system. As shown in rebuttal Fig.14a, glucose deprivation-resistant H292 cells (H292-V) had citrate synthase levels similar to parental H292 cells. In addition, ectopic expression of KRas^{G12V} or berberine treatment had little effect on citrate synthase expression (Rebuttal Fig.14b-c). These results indicate that glucose deprivation, KRas^{G12V}, or berberine had minimal impact on mitochondrial mass. Accordingly, we have replaced the Mitotracker analyses with citrate synthase abundance. These new results were integrated into the revised Fig. S1g, S9b, and S11c.

Rebuttal Fig.14. Glucose deprivation, KRas^{G12V}, or berberine has minimal impact on mitochondrial mass.

Point 4:

- It is not specified whether authors measure total FAD levels. If so, authors should also measure FAD⁺:FADH₂ ratio. Similarly, it is not clear which form of FAD they provide to the cells in supplementation assays. As a side note, I was not aware of any FAD transporters on the plasma membrane; authors may want to briefly discuss how FAD enters the cell.

Response 4: Thank you for the comments and suggestions. We used an FAD assay kit (Abcam, ab204710) to examine cellular FAD levels. This kit specifically measures FAD⁺ levels but does not detect FADH₂ levels. In our supplementation assays, we treated the cells with FAD⁺ (HY-B1654A, MCE) rather than FADH₂. Moreover, detecting the ratio of FAD⁺ to FADH₂ is technically unfeasible due to their similar chemical properties. We have clarified these points in the revised manuscript.

Currently, there are no reports indicating the existence of an FAD transporter. Yet, our results show that treatment with FAD (dissolved in DMSO) significantly increased intracellular FAD levels in a dose-dependent manner (Rebuttal Fig.15a), in keeping with a previous report (*Hirano et al., FEBS Open Bio., 2017, PMID: 29226080*). Notably, FAD supplement could markedly restore cellular FAD levels and c-JUN/GLS1 protein expression, both inhibited by silencing of MT-CO2 (Rebuttal Fig.15b-c). We further demonstrate the vital role of FAD in c-JUN/GLS1 expression by supplementing a precursor of FAD, riboflavin, which can be transported into the cell by its specific transporter (Rebuttal Fig.15d-f). These new results were integrated into the revised Fig.3i-o and S4n-o.

Rebuttal Fig.15. FAD (a-c) or riboflavin (d-f) treatment could markedly restore both cellular FAD levels and the c-JUN/GLS1 protein expression inhibited by silencing of MT-CO2.

Point 5:

- Authors should take a deeper look at the effects of α-KG supplementation

Response 5: Thank you for the suggestion. Accordingly, we performed new experiments by supplementation with α-KG. Our results show that α-KG could rescue tumor cell viability inhibited by the silencing of MT-CO2 under Glc-/Gln condition (Revised Fig.2h-i), whereas it had little effect on the levels of H3K9me2 and H3K4me2 upregulated by silencing of MT-CO2 (Rebuttal Fig.11), indicating that α-KG serves as an intermediate of glutaminolysis to rescue tumor cell survival upon glucose deprivation and that α-KG does not impact MT-CO2-mediated epigenetic alternation.

Point 6:

- Authors draw conclusions on the role of TFAM in mediating KRAS signaling, but no experiments actually interrogate TFAM. Please, either show experiments or re-state conclusions.

Response 6: Thank you for the comment and suggestion. Our results in the previous manuscript showed that activated Ras significantly upregulated TFAM mRNA and protein levels (Rebuttal Fig.16a-b). Silencing of TFAM dramatically inhibited Ras-induced upregulation of MT-CO2, c-JUN, and GLS1 expression,

accompanied by reduced FAD levels (Rebuttal Fig.16c-e). Importantly, KRas^{G12V}-induced cell survival in response to glucose starvation was significantly inhibited by the knockdown of TFAM (Rebuttal Fig.16f). Together, these results indicate that oncogenic Ras upregulates the expression of TFAM to promote MT-CO2 transcription. These results were integrated into the revised Fig.S9d-i.

Rebuttal Fig.16. Oncogenic Ras upregulates the expression of TFAM to promote MT-CO2 transcription.

Point 7:

- Please, show experiments with acute MEK inhibition. Do those alter mt-CO2 at shorter time points? An argument against shRNA-type experiments is that cells are metabolically flexible over time and some rearrangements might be secondary to adaptation rather than direct signaling effectors.

Response 7: Thank you for the comment and suggestion. Accordingly, we treated H1299 cells with the MEK inhibitor PD98059. As shown in rebuttal Fig.17, PD98059 markedly suppressed MT-CO2 expression in a time-dependent manner. This new result was integrated into the revised Figure S9j.

Rebuttal Fig.17. PD98059 suppresses MT-CO2 expression.

Point 8:

- LSD1 is believed to act primarily on H3K4me2 and only marginally on H3K9me2 (if anything at all). While an effect by a related histone de-methylase is plausible, authors should show data on H3K4me2 if they focus on LSD1. Please, repeat ChIP experiments w/ H3K4me2.

Response 8: Thank you for the comment and suggestion. Accordingly, we examined the impact of MT-CO2 knockdown on H3K4me2 levels and its binding to the *JUN* promoter. As shown in Rebuttal Fig. 18a-b and revised Fig. 3q, MT-CO2 silencing increased H3K4me2 levels, and H3K4me2 was found to bind to the *JUN* promoter. However, our ChIP analyses revealed that MT-CO2 knockdown had minimal impact on H3K4me2 binding affinity at the *JUN* promoter (Rebuttal Fig. 18c). Therefore, these results further support the conclusion that MT-CO2 regulates *JUN* transcription via the FAD-LSD1-H3K9me2 axis. These new results were integrated into the revised Fig. S5d-f.

Rebuttal Fig.18. (a) H3K4me2 could bind to the JUN gene promoter region in A549, HepG2, or HUVEC cells. Histone Modifications dataset from ENCODE/Broad Institute was used for this analysis. (b-c) Silencing of MT-CO2 has little effect on H3K4me2 binding to the *JUN* gene promoter.

Point 9:

- The experiments using berberine, and the conclusions authors draw are highly perplexing. Berberine is a recognized Complex I inhibitor, so parsing out the effects on Complex I vs Complex IV is hard, if not impossible. More in general, target specificity is a concern. Mitochondrial mass is measured only using Mitotracker, and that must be fixed. Rescue with FAD is not performed (rather, authors choose to rescue with a-KG for some reasons). Tumor data are OK, but authors should also show how treatment affects *Kras*WT tumors (to confirm specificity for *Kras*MUT tumors). To be honest, I thought this part did not add anything to the manuscript but rather undermined a little bit its overall value.

Response 9: Thank you for the comments and suggestions. Although berberine is a known complex I inhibitor, our results show that berberine downregulated MT-CO2 expression via suppressing TFAM expression,

suggesting that berberine can also suppress the activity of complex IV. Notably, metformin, a documented Complex I inhibitor, can also inhibit complex IV activity (LaMoia et al., PNAS, 2022, PMID: 35238637).

As suggested, we show that berberine had little effect on citrate synthase expression (rebuttal Fig.14c), indicating that berberine had little impact on mitochondrial mass (revised Fig.S11c). Since our results show that berberine down-regulates MT-CO2 to inhibit GLS1 expression and glutaminolysis, consequently suppressing tumor cell viability upon glucose deprivation, we reasoned that the direct rescuing experiment should use an intermediate downstream of GLS1. Therefore, we used α -KG to rescue berberine-mediated cell death under glucose deprivation (Revised Fig.7b and 7d).

Regarding the effects of berberine on Ras wild-type tumor cell growth, our new results show that a low dose of berberine (0-1.0 μ M) significantly suppressed the colony formation of RAS mutant tumor cells (A549, KRas^{G12S}) while having little effect on RAS wild-type tumor cells (PC-9) (Rebuttal Fig.19a-b), suggesting that mutant RAS-driven tumor growth is more susceptible to berberine treatment. Further investigation shows that berberine could inhibit the LLC (Lewis lung carcinoma cells harboring KRas^{G12C})-derived tumor growth and prolong the overall survival in immune-competent mice (C57BL/6J) (Rebuttal Fig.19c-f). Together, these results highlight berberine as a promising drug for treating Ras-activated cancers. These new results were integrated into the revised Fig.7m-o, S11n, and S12.

Rebuttal Fig.19. Berberine treatment suppresses the proliferation of RAS mutant tumor cell proliferation in vitro and tumor growth in vivo.

Minor issues:

Point 10:

- in Fig 1, one mitochondrial gene (*mtATP8*) is missing.

Response 10: Thank you for pointing out this issue. Accordingly, we repeated qPCR analyses to examine MT-ATP8 expression. As shown in rebuttal Fig.20, glucose deprivation-resistant H292 cells (H292-V) had similar MT-ATP8 mRNA levels compared to parental H292 cells. This new result was integrated into the revised Fig. 1g.

Rebuttal Fig.20. H292-V cells have similar MT-ATP8 mRNA levels compared to parental H292 cells.

Point 11:

- some terminology is funky, like "mass spectrum" or "transcriptional factors"

Response 11: We appreciate your comments regarding the terminology. We have carefully reviewed the revised manuscript and ensured the correct use of terminology, including "mass spectrometry" and "transcription factors."

Point 12:

- Along the same point, the saying "H3K9me2 expression" is misleading as an histone mark is not technically expressed but a post-translational modification. Please, refer to H3K9me2 levels.

Response 12: Thank you for pointing out this issue. We have used "H3K9me2 levels" in the revised manuscript to ensure clarity and accuracy.

Point 13:

- For RNA-Seq in Fig 3A, I would have appreciated a more unbiased approach

to data analysis.

Response 12: Thank you for your comment. we selected the ten most significantly changed transcription factors affected by the silencing of MT-CO2.

Reviewer#3 ' s comments about Reviewer#1 ' s previous comments:

The first point that Reviewer#1 posed is a major conceptual roadblock. Authors articulate their response very well, supporting their hypothesis with several new experiments. The hypothesis sounds a bit far-fetched but results hold their own. Of course, this introduces a chapter that was not present in the original manuscript and would probably require a more in-depth revision from qualified people. For what I can say, results are interesting and look legit.

The point on FAD is well taken by Reviewer#1. I also have my concerns. Authors did answer and results are OK, yet FAD role still sounds puzzling. Minor: I think there is an error in the labelling of shRNA used in Reb Fig 7e (it should be sh-MT-CO2).

Reviewer#1 was critical of introduction and discussion. Authors did not change much in the manuscript but replied in detail. I tend to agree with Reviewer#1, but I don ' t see deal breakers.

Authors tackled the issue(s) with GLS ' s role

Response: We thank the reviewers for their insightful comments.

With the issue of the labeling of shRNA used in Reb Fig 7e, we investigated the effect of berberine, rather than MT-CO2 silencing by shRNA, on KRas^{G12V}-driven tumor growth *in vivo*. Therefore, the label was berberine (Ber), but not shMT-CO2.

As the issue highlighted by the previous Reviewer #1, "*the relevance and significance of glutaminolysis in vivo, along with the clinical failure of GLS1 inhibitors, were not adequately addressed*". In response, we have incorporated a new paragraph in the revised discussion (Lines 579-589).

We are grateful for the reviewer ' s valuable feedback and hope the revisions are satisfactory.